

# Local soil quality assessment of north-central Namibia: integrating farmers' and technical knowledge

Brice Prudat[1], Lena Bloemertz[1], Nikolaus. J. Kuhn[1]

[1]Physical Geography and Environmental Change, University of Basel, Basel, 4000, Switzerland

*Correspondence to*: Brice Prudat (brice.prudat@unibas.ch)

**Abstract.** Soil degradation is a major threat for farmers of semi-arid north-central Namibia. Soil conservation practices can be promoted by the development of soil quality (SQ) evaluation toolboxes that provide ways to evaluate soil degradation. However, such toolboxes must be adapted to local conditions to reach farmers. Based on qualitative (interviews and soil descriptions) and quantitative (laboratory analyses) data, we developed a set of SQ indicators relevant for our study area that integrate farmers' field experiences (FFE) and technical knowledge. We suggest using participatory mapping to delineate soil units (Oshikwanyama Soil Units, KwSUs) based on FFE, which highlight mostly soil properties that integrate long-term productivity and soil hydrological characteristics (i.e. *internal SQ*). The actual SQ of a location depends on the KwSU described and is thereafter assessed by field soil texture evaluation (i.e. *chemical fertility potential*) and by soil colour shade (i.e. *SOC status*). The resulting information includes *internal SQ* (KwSU), *chemical fertility potential* (sand content) and the *soil organic carbon content status* (colour shade). This three-level information reveals SQ improvement potential and aims to help farmers, rural development planners and researchers from all fields of studies understanding SQ issues in north-central Namibia. This SQ toolbox is adapted to a restricted area of north-central Namibia but similar tools could be developed in most areas where small-scale agriculture prevails.





**1    Introduction**
Soil degradation is a major cause of marginal agricultural productivity and food insecurity in sub-Saharan Africa
(FAO and ITPS, 2015). In north-central Namibia (NCN), increasing land tenure security through the Communal
Land Reform Act of 2002 (Government of the Republic of Namibia, 2002) aims to increase investment in land
and improve soil quality (SQ) in communal areas (Adams et al., 1999). The state of environmental and soil
degradation remains, however, unclear in the area (Newsham and Thomas, 2011). The selection of SQ indicators
adapted to local conditions thus represents an important step towards sustainable soil management practices
(Ditzler and Tugel, 2002). We consider that the SQ is a function of soil properties, intended land use and
management possibilities and goals (Andrews et al., 2004). This definition favours a use-dependent approach,
which is in line with farmers' and local administration's needs. A bottom-up approach is vital as farmers are the
key actors for developing and implementing soil management policy (Mairura et al., 2007).
**1.1    Technical soil quality assessment**
Many SQ indicators have been developed over the past decades (e.g. Mueller et al., 2010; Wienhold et al., 2004)
and the need to adapt SQ indicators to local conditions was acknowledged very early (Granatstein and Bezdicek,
1992; Nicholls et al., 2004). Most of the indicators require measuring physical, chemical and/or biological soil
characteristics that need laboratory measurements, specific technical material and/or experts' knowledge (Table
1). Therefore, most SQ indicators cannot be used directly by farmers (Nicholls et al., 2004), which is particularly
problematic in low-income regions due to limited availability of laboratory and experts' services (Musinguzi et
al., 2015), like in NCN.
Many SQ indicators are based on yield data collected during two (e.g. Andrews et al., 2004) or even only one
year (Hillyer et al., 2006). With such short records, it is impossible to consider how inter-annual climatic
variability affects subsistence farmers, who aim to reduce the risk of harvest failure (Graef and Haigis, 2001).
Therefore, most SQ indicators developed using yield data collected during periods too short to fully reflect
climatic constraints to production are of limited relevance in areas with high interannual rainfall variability.
Considering the shortcomings of some SQ indicators, it is therefore imperative to develop "cost-effective and
user-friendly tools" (Musinguzi et al., 2015) to evaluate SQ based on land users requirements.



Table 1: Frequently used soil properties that may be used as field soil quality (SQ) indicators, possible field
measurements techniques and challenges for local users (adapted from Wienhold et al., 2004)).

| Soil properties | Field measurements | Challenges for local use |
|---|---|---|
| **Physical** | | |
| Texture | Texture-by-feel<br>Kruedener test | Subjectivity, expert knowledge<br>Specific material |
| Depth of topsoil | Observation | Expert knowledge |
| Bulk density | Weighing scale | Dry soil required, specific material |
| Infiltration | Infiltrometer | Time consuming, specific material |
| Water holding capacity | Estimation from texture | Subjectivity, Specific material (see above) |
| **Chemical** | | |
| Organic C | Estimation from colour | Approximation, Colour chart |
| Total N | Test kit | Specific material |
| pH | pH-Hellige | Specific material |
| Electrical conductivity | Probe, sensors | Specific material |
| Extractable N, P, K | Test kit | Specific material |
| **Biological** | | |
| Microbial biomass C and N | Unknown | |
| Potentially mineralisable-N | Test kit | Specific material |
| Soil respiration | Test kit | Specific material |

## 1.2   **Farmers' field experiences**
Farmers' field experiences (FFE) include all farmer-based soil fertility assessment techniques (Musinguzi et al.,
2015). This terminology is preferred over "indigenous knowledge" or "local knowledge" because it refers to a
clearly defined group of land users, all people involved in farming (farm owners, workers, children). FFE are
essential as entry point for outsiders to understand local land use practices and local soil diversity (Mairura et al.,
2007; Ramisch, 2004). Many studies incorporate FFE to select the most appropriate properties to use as SQ
indicators (Musinguzi et al., 2015; Nicholls et al., 2004). The resulting local SQ indicators cover broader
agronomic properties than technical SQ indicators as they may account for economic issues (Warren, 1991),
long-term productivity or risk management practices (Graef and Haigis, 2001) , for example dealing with rainfall
variability.
Aside from improving the relevance of SQ indicators, the use of FFE involves farmers in the process of
agricultural evolution (Ditzler and Tugel, 2002; Mairura et al., 2007; Warren, 1991). However, FFE can be
inaccurate, biased by social context (Gray and Morant, 2003) and resilient against environmental and socio-
economic changes (Briggs and Moyo, 2012). Technical knowledge, on the other hand, is valuable for its level of
standardisation, which allows for spatial and temporal comparisons and facilitates international communication
(Niemeijer and Mazzucato, 2003). Scientists should therefore integrate both knowledge systems to provide tools
connecting FFE and technical knowledge (Lima et al., 2011). Methodologies to select indicators for SQ based on
the integration of FFE with technical knowledge have been developed, discussed and yielded promising results
(Barrios et al., 2006). Most studies concerning integrated soil knowledge showed the parallels between farmers'





and technical assessment, but only a few developed local SQ toolboxes to fully evaluate the SQ conditions
(Table 2).
Table 2: Selection of studies suggesting series of local SQ indicators. SOM= Soil Organic Matter, SOC= Soil
Organic Carbon.

| References | Local Soil Quality Indicators | Toolbox for SQ evaluation |
|---|---|---|
| Ditzler and Tugel (2002) | Compaction, drainage/infiltration, nutrient-holding capacity, salinity, soil organisms, earthworms, residue decomposition, crop vigour. | Farmers' evaluation; Qualitative and subjective evaluation |
| Gruver and Weil (2007) | SOM, crop performance, soil water availability, erosion history. | Soil C and structure evaluation; No method suggested |
| Lima et al. (2011) | Earthworms, soil colour, yield, spontaneous vegetation, SOM, root development, soil friability, rice plant development. | No method suggested |
| Mairura et al. (2007) | Crop yield, soil colour, texture and tilth, soil macro fauna, the abundance or diversity of weed species. | No method suggested |
| Murage et al. (2000) | Crop performance, soil tilth, moisture and colour, presence of weeds and soil invertebrates. | SOM or KMnO4 oxidisable C; Laboratory measurements |
| Musinguzi et al. (2015) | | FFE and scientific quantitative rating with SOC; Laboratory measurement |
| Nicholls et al. (2004) | Structure, compaction, soil depth, status of residues, colour, odour, SOM, water retention, soil cover, erosion, presence of invertebrates, microbiological activity**.** | Farmers' evaluation; Qualitative and subjective evaluation |

Farmers knowledge of environmental factors and SQ in NCN has been already collected and discussed in
various studies (Hillyer et al., 2006; Rigourd et al., 1999; Verlinden and Dayot, 2005), but there is still "a lack of
understanding [of local land classification system] by scientists or extensionists […]" (Verlinden and Dayot,
2005). A relatively high number of "indigenous land units" were described based on vegetation, landforms
and/or soils (Hillyer et al., 2006; Verlinden and Dayot, 2005a). These studies present an interesting collection of
FFE, but none was developed into locally adapted SQ indicators. Yet, such indicators are essential to allow
researchers and farmers to assess SQ at a specific location and time-period relevant for agricultural cycles
(Barrios et al., 2006).
Based on qualitative (semi-structured interviews, soil profile descriptions) and quantitative data (field soil profile
descriptions, laboratory measurements), we aim to develop a set of SQ indicators relevant for our study area that
integrate FFE and technical assessment. Following Barrios and Coutinho (2012) these indicators must be: a)
Practical and easy to use under field conditions; b) easy to interpret; c) relatively economical; d) sufficiently
sensitive to highlight the changes under study; e) integrate physical, chemical and biological characteristics and
processes; f) useful for estimating all relevant soil properties; g) give good correlations between plant
productivity and soil health. We aim to verify the benefits of using FFE for soil quality assessment as the
development of SQ estimation tools is vital for SQ management in areas where small-scale family agriculture
represents large proportion of land use.



**2    Methods**
2.1   **Study area**
In NCN, the climate is semi-arid subtropical with a rainy season from December to April. Average annual
precipitation ranges from 350 to 550 mm with large inter- and intra-annual variability (Mendelsohn et al., 2000).
In Ondangwa, the annual rainfall during 1959–1973 ranged from 200 to 1039 mm with an average of 495 mm
(Verlinden et al., 2006). Crop production failure because of rain quantity and distribution occurs every second
year (Keyler, 1995). The area lies over the Owambo sedimentary basin with the upper part constituted of aeolian
sands redistributed throughout the Quaternary Period (Miller et al., 2010). The region is characterised by the
endorheic Cuvelai drainage basin and the north-eastern Kalahari woodlands or Kalahari Sandveld (Figure 1;
Mendelsohn et al., 2000).

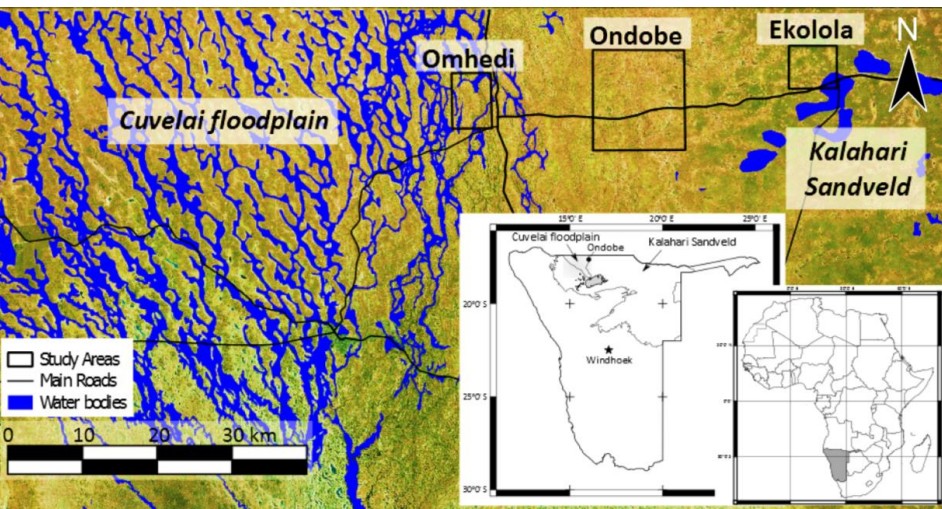

Figure 1: Overview of Southern Africa and satellite view (enhanced colour saturation) of north-central Namibia
with the Cuvelai floodplain (north-west), the Kalahari Sandveld (north-east) and location of the three study areas
(Omhedi, Ondobe and Ekolola). Vegetation appears in green, bare soil appears in orange, water bodies in blue.
Non-commercial agricultural activities are the most important land use in NCN (Mendelsohn et al., 2000).
Around 120'000 households are farming in the region, mostly cultivating small-scale (1-4 ha) rainfed pearl
millet (*Pennisetum glaucum*; Mendelsohn et al., 2013). Average yields of millet are very low, (220 kg ha$^{-1}$ in
average in Ohangwena region), highly variable from year to year and from household to household, due to low
soil fertility, low nutrient supply, irregular rainfall and pests (Central Bureau of Statistics, 2003; Mendelsohn et
al., 2000; Rukandema et al., 2009).
Three groups of villages in Ohangwena region were selected (Omhedi, Ondobe, Ekolola; Figure 1) based on
dialect homogeneity (Oshikwanyama) and environmental heterogeneity (vegetation, soils). These villages lie on
a west-east climatic, edaphic and land-use gradient with a mosaic pattern of soil and vegetation (Mendelsohn et
al., 2013). The annual rainfall quantity, the proportion of deep sandy soils and forest cover increase eastwards.
The westernmost area (Omhedi) is largely influenced by the active drainage system of the Cuvelai River, which
creates a network of water channels (called locally *iishana*) that significantly influenced soil development





(fluvial deposits, salinization). Ondobe is located between the drainage basin in the west, and the Kalahari
Sandveld in the east. Further east, Ekolola is characterised by the Kalahari Sandveld , which is dominated by
deep loose sand deposits (Mendelsohn et al., 2000). All three areas were recently settled by immigrants from
Angola, mostly during the 1910-1920s', but population density increased more dramatically in the westernmost
areas due to water accessibility (Kreike, 2004).
## 2.2    Assessment of farmers' field experiences
From February 2013 to June 2014, 46 farms were visited, in which 87 semi-structured interviews were
conducted to collect FFE, mainly in Ondobe (52 interviews held in 22 farms). Some farmers were visited several
times. Mostly people above the age of 50 (75 % of interview time) were surveyed because of their availability to
talk and the knowledge they wished to share, typically elderly men (49 % of total interview time). Most
interviews were held in the house providing conceptual references, but some were held in the fields or in front of
soil pits, providing locational references (Oudwater and Martin, 2003). Questions aimed to generate information
on the types of soil that are cultivated and the characteristics that differentiate them. With "Oshikwanyama Soil
Units" (KwSU) we refer to the soil units that are distinguished by the farmers by sight, touch, experienced yields
or others (following the definition of Indigenous Land Units suggested by Verlinden and Dayot (2005).
All the interviews were held in *Oshikwanyama* and audio-recorded. Direct interpretation was performed by,
mostly, Ms Martha Shekupe Fillemon (20). The English interpretation was afterwards completely transcribed.
Parts of the interviews were transcribed in *Oshikwanyama* and translated into English by non-professional local
translators. The interviews were annotated using MaxQDA 11 (VERBI GmbH, 2014) to facilitate the qualitative
data analysis. The annotation system included KwSU names (*omutunda*, *omufitu*, *elondo*, *ehenene*, *ehenge*) and
"soil quality". The latter annotation was used to select quotes in which a certain location or a specific KwSU was
characterised with regards to the suitability for pearl millet cultivation.
Over the total number of informants (46), we calculated the proportion of them who mentioned each KwSU.
Afterwards we associated these interviews to specific soil properties, which are finally grouped into five
frequently mentioned properties: hardness, soil hydrology, productivity potential, soil colour shade and soil
colour hue.
## 2.3    Technical knowledge collection
In cultivated fields, 29 soil profiles were described, mostly in Ondobe (n= 22), but also in Omhedi (n= 3) and
Ekolola (n= 4). The 29 soil profiles were classified as *omutunda* (n= 15), *ehenge* (n= 4), *omufitu* (n= 4), *elondo*
(n= 3) or *ehenene* (n= 3) by the farmers. For the analysis, we concentrated on *omutunda* given its high
agricultural value and its prevalence in the cultivated area.
### 2.3.1    Field soil profile description and sampling
The *Guidelines for soil description* (FAO, Land and Water Division, 2006) were used for standardised soil
profile description. In the context of this study, we only discuss the horizon limits, clods consistence, bulk
density and moist colour down to 40 cm, as they are best suited to the objective of developing an SQ tool that
could be used by various land users, who have not the resources and expertise to go through a full soil



description. Soil colour was estimated in the field using Munsell soil colour chart on a moist sample for each
horizon. Soil colour provides information about soil formation processes (e.g. leaching, clay alteration) and soil
organic carbon content (SOC) (Viscarra Rossel et al., 2006). The dry consistence was evaluated by crushing a
clod of soil between the fingers. This property informs on the amount and type of clay, SOC and soil particles
organisation (FAO, Land and Water Division, 2006).
Two sample rings were collected from each described horizon and homogenised to create a single mixed sample
per horizon. Dried-samples were then sieved (2 mm) and used for further analysis.

### 2.3.2    Laboratory analyses

Soil texture is the most important soil characteristic having a direct influence on most soil processes and
properties (Vos et al., 2016). It was calculated using laser diffraction (Malvern Mastersizer 2000) that measures
volumetric particle-size distribution. Prior to measurement, samples were shaken overnight in water and
dispersed during 300 s with 30 mW ml$^{-1}$ (9 J ml$^{-1}$) ultrasonic energy. The particle size class <20 μm was
considered as the active mineral fraction (Feng et al., 2013).
SOC plays an important function as adsorbing material and is often used to evaluate SQ (Musinguzi et al., 2015).
SOC saturation (C-saturation) is defined as "the ratio of the present topsoil total [SOC] level relative to the same
soil in its undisturbed […] state" (Sanchez et al., 2003). Various models have been developed to evaluate the
SOC of a C-saturated soil (Six et al., 2002; Zinn et al., 2007), for example based on the proportion of <20 μm
fraction (Feng et al., 2013). We choose the model from Feng *et al.* (2013) because it is based on a large review
of studies, and developed for soils with predominantly 1:1 clay minerals, common in the tropics.
SOC and inorganic carbon contents were determined with a LECO® analyser (RC-612). Soil electrical
conductivity and soil solution pH were measured in 1:5 (soil—water) suspension and pH$_{CaCl2}$ in a 1:5 (soil—0.01
M CaCl$_2$).
Cation exchange capacity and base saturation values indicate the cation reservoir of a soil and are important
characteristics to evaluate the ability of a soil to sustain plant growth. Both properties were not measured in this
study because the presence of calcium carbonates and soluble salt strongly influences the measurements (Sparks
et al., 1996), which makes results very difficult to interpret, especially that the expected values were very low.
Instead, we used robust and sufficiently accurate methods as proxy for cation exchange capacity (soil organic
carbon and the <20 μm fraction content) and for base saturation (soil pH)(Blume et al., 2011).

### 3    Results and Discussion

#### 3.1    Oshikwanyama Soil Units: A homogeneous body of soil knowledge

Like in many areas worldwide (Barrera-Bassols and Zinck, 2003), farmers of NCN classify soil potential (mostly
with regards to pearl millet cultivation) using several properties. In cultivated areas, five KwSUs were frequently
described: *omutunda*, *ehenge*, *ehenene*, *omufitu*, and *elondo* (Table 3). Knowledge and descriptions of these local
soils were largely shared among the interviewed population, and we did not observe differences based on gender,
generations or studied eco-regions. Some criteria used in the FFE were general (e.g. productivity potential),
while others were more specific (e.g. soil colour shade and hardness, waterlogging risk; Table 3).





KwSUs' names define specific objects in the landscape. For example, the suffix –*tunda* in *omutunda* means
"something on a hill" (TN, 65, Ekolola)[1] and *omufitu* refers to woodlands located close to villages ("a land with
many bushes and trees"; KS, 60, Ondobe). These names are instilled in the everyday language, which explains
the homogeneity of the soil-related vocabulary among the population and suggests that labelling of places (with
KwSUs) changes little over time.
We calculated the proportion of informants mentioning specific characteristics for each KwSU to highlight the
most prominent characteristics, per KwSU and based on total number of informants mentioning any of the five
KwSUs (Table 3). The properties that were the most frequently used to describe KwSUs were related to soil
hardness (63.5 %), productivity potential (57.7 %), soil hydrology (43.8 %) and soil colour shade (38.0 %). The
morphological properties (colour shade, consistence) referred mostly to topsoil layers as farmers indicated
characteristics that were discussed during transect walks. As observed by Verlinden and Dayot (2005) for the
Indigenous Land Units, the predominance of each characteristic varies depending on the KwSU. For example,
hardness/softness is a prominent characteristic to describe *omutunda* and *omufitu* (used by resp. 72.2 % and
70.8 %) while soil hydrological characteristics were important to describe *ehenge* (68.8 %).
Table 3: List of farmers' field experiences characteristics used to describe each KwSU, with the number of
informants mentioning each KwSU (*n*) and the proportion of informants mentioning each characteristic (in
relation to *n*). Values are only indicative as the data collection method was not adapted for statistical analyses.

| KwSUs | Number of informants mentioning the KwSU (*n*= 46) | Hardness/ Softness (in %) | Soil hydrology (in %) | Productivity potential (in %) | Soil colour shade (in %) | Soil colour hue (in %) |
|---|---|---|---|---|---|---|
| *Omutunda* | 36 | 72.2 | 36.1 | 66.7 | 33.3 | 0.0 |
| *Omufitu* | 24 | 70.8 | 41.7 | 66.7 | 50.0 | 12.5 |
| *Ehenge* | 32 | 53.1 | 68.8 | 43.8 | 25.0 | 6.3 |
| *Ehenene* | 29 | 62.1 | 51.7 | 51.7 | 41.4 | 0.0 |
| *Elondo* | 16 | 56.3 | 0 | 62.5 | 50.0 | 56.3 |
| Average | | 63.5 | 43.8 | 57.7 | 38.0 | 10.2 |

The high frequency of interviews mentioning productivity (57.7%; Table 3) might have been influenced by the
aim of the study and frequent questions concerning productivity by the researchers. Farmers considered
unanimously *omutunda* as the most fertile soil and agreed that pearl millet productivity is strongly limited in
*ehenene* (Table 4). Productivity in *elondo*, *ehenge* and *omufitu* did not reach consensus. The productivity of these
KwSUs may largely depend on factors less dependent on soil (rainfall, fertiliser availability). Notably, *ehenge* is
good in poor rainfall years, but poor in good rainfall years (Table 4).

---

[1] To keep the informants anonymous, we used a code that indicates: 1) a two-letter name, 2) the farmers' age and 3) the study area of the farm.




The productivity of soils depends not only on internal soil properties and processes (waterlogging risks,
landscape position) or climatic conditions, but also on management strategies (e.g. fertiliser application). The
effect of management was acknowledged by farmers who explained that KwSUs represent not accurately the
actual SQ (e.g. "*omutunda* is not always fertile, it needs to be dark", SK, 60, Ondobe). Farmers estimated the
actual SQ of a location also based on crop health, soil consistence and soil colour shades ("[...] needs to be
dark", SK, Ondobe), hardness ("millet likes hard soil", HP Ondobe). We will discuss the technical significance
of these properties below.
Table 4: List of the KwSUs identified and the most-frequently used farmer's field experiences characteristics.
GRY= good rainfall year; PRY= poor rainfall year.

| | Soil type attributes and local soil indicator | | | Suitability for pearl millet |
|---|---|---|---|---|
| | **Soil hydrology** | **Consistence** | **Colour shade** | |
| *Omutunda* | No waterlogging High water retention Dries out quickly | Hard | Dark/black | GRY: Very good PRY: Strongly reduced |
| *Omufitu* | No waterlogging Low water retention | Loose | Dark or light | GRY: Poor PRY: Poor |
| *Elondo* | No waterlogging | Intermediate | Intermediate | GRY: Good PRY: ND |
| *Ehenge* | Waterlogging risk Dries out very slowly | Loose | Light/white | GRY: Poor PRY: Good |
| *Ehenene* | Waterlogging risk Low water retention Dries out quickly | Hard | Light/white | GRY: Very poor PRY: Very poor |

Soil hydrological properties were mentioned frequently to describe KwSUs. These properties need to be
understood in relation to rainfall variability (Table 4). Productivity of *omutunda* drops during droughts ("pearl
millet is burned", JL, Ondobe), while it increases in *ehenge* ("*ehenge* is good in year with lack of rain", LS,
Ondobe). Therefore *ehenge* secures minimum harvest during poor rainfall years, which is essential for farmers
relying on yearly food production (Graef and Haigis, 2001). Conversely, *ehenge* undergoes waterlogging during
good rainfall years ("[*ehenge*] used to be full of water", NJ, Ondobe), which strongly limits pearl millet growth.
These soil hydrological characteristics are difficult to assess during standard field surveys and the integration of
these characteristics in KwSU definitions is crucial for SQ evaluation as soil water availability is the most
significant limitation in semi-arid regions (McDonagh and Hillyer, 2003).
**3.2    Technical analysis of farmers' field experiences**
Results from technical analyses are summarised in Table 5. In this table, the soil characteristics are calculated for
the layer 5-15 cm and 25-35 cm using an arithmetic mean of the different values weighted by the depth of each
horizon. All described soils have very low organic carbon (<5 mg C g$^{-1}$ soil) and high sand content (>70% in the
5-15 cm layer). *Omutunda* has a larger proportion of <20 µm fraction (6.5 to 22.8% in the layer 5-15 cm) and
more SOC (1.4 to 4.4 mg C g$^{-1}$ soil) than all other studied KwSUs. Furthermore, slightly alkaline conditions



(Table 5) indicate a high base saturation. All these characteristics suggest high chemical exchange capacity in
*omutunda* and chemical fertility. A slightly more acid soil solution, a smaller amount of <20 μm particles and
SOC in *elondo* indicate lower chemical fertility. The proportion of <20 μm fraction in *ehenene* can be high (up
to 16.4 %), but very high pH (8.4 to 10.1 in the 25-35 cm layer, increasing with depth) restricts plant growth. All
*ehenge* and *omufitu* described have very low proportion of <20 μm fraction (<6.5 %) down to 40 cm. Our
laboratory results therefore support farmers' assessment pointing to the greater chemical fertility potential of
*omutunda*.
### 3.3 *Omutunda*: uniform or plural?
FFE and our technical analyses indicate a large diversity in *omutunda* soils. The diversity is expressed in FFE, as
not all *omutunda* are similar and as their productivity varies ("The soil [*omutunda*] … inside the country
[floodplain] breastfeeds on water streams … it is hard not like ours", TN, 70, Ekolola). Technical analyses
support this observation as various measured properties show a large coefficient of variation (Table 6), like the
proportion of <20 μm fraction, pH (CaCl$_2$) and Munsell colour values (Figure 2), especially in the topsoil. Given
the high proportion of *omutunda* described in Ondobe (n= 10) in comparison to the other areas (Omhedi= 2,
Ekolola= 3), the statistics presented in Table 6 are skewed towards the characteristics of *omutunda* in Ondobe.
This does not jeopardize the substance of these results given the diversity found in the area (transition from
floodplain environment to Kalahari woodlands).
From FFE perspective, *omutunda* was mostly defined by excluding areas not suitable for pearl millet because (i)
it does not experience waterlogging (hypoxic conditions); (ii) it does not have loose sand topsoil (very poor
chemical fertility); and (iii) it does not have very shallow fragipan (limits water storage capacity and restrict
workability). Pearl millet can be cultivated on various soils (Baligar and Fageria, 2007), which contributes to the
large variability of soils considered as suitable for its cultivation. Temporal variation of SQ was acknowledged in
FFE and various degrees of degradation (e.g. organic and nutrients depletion, salinization) lead to variability in
SQ of *omutunda* at a specific time. Management practices (amount of fertiliser, ploughing) therefore also
contribute to add some variability. There were small differences depending on the area of study and the
surrounding environment (Table 5). *Omutunda* described in Ekolola (Kalahari Sandveld environment) has
coarser texture compared to the *omutunda* described in Omhedi and Ondobe (Floodplain; Table 5). These
differences were expected as FFE were constructed based on comparative observations (e.g. "harder than") and
therefore influenced by the surrounding environment (Birmingham, 2003; Niemeijer and Mazzucato, 2003).
The variability described in the various studied *omutunda* illustrates the need of developing tools for
standardisation. This would help avoiding classifying soils that should not be compared directly, but need to be
considered as various entity that show similar features.



1  Table 5: Chemical and physical characteristics of topsoil (5-15 cm) and subsoil (25-35 cm) layers of the studied
2  soil profiles. Quantitative data are represented by average values and Colour Hue by the most frequent value.

| KwSU | Profile | Area | TOC (mg C g⁻¹) Top | TOC Sub | < 20 µm (%) Top | < 20 µm Sub | Sand (%) Top | Sand Sub | pH (CaCl₂) Top | pH Sub | EC (µS m⁻¹) Top | EC Sub | Moist colour Top | Moist colour Sub | SQ evaluation |
|---|---|---|---|---|---|---|---|---|---|---|---|---|---|---|---|
| Ehenene | EFIDI_01 | Ondobe | 0.7 | 0.6 | 2 | 97 | 5.7 | 93.5 | 8.08 | NA | 2.3 | 7.2 | 2.5Y 7/3 | 2.5Y 8/3 | Ehenene Very poor - |
| Ehenene | NDOB_02 | Ondobe | 1.6 | 1.8 | 18.1 | 80.6 | 16.1 | 82.8 | 6.51 | NA | 2.4 | 3.7 | 10YR 4/2 | 10YR 3/2 | Ehenene Very poor 0 |
| Ehenene | NDOB_18 | Ondobe | 1.3 | 0.8 | 3.2 | 94.9 | 17.5 | 80.7 | 6.12 | 7.82 | 0.4 | 1.1 | 2.5Y 5/2 | 2.5Y 6/2 | Ehenene Very poor + |
| Ehenge | EFIDI_02 | Ondobe | 1 | 0.6 | 2.6 | 97.4 | 4.1 | 95.2 | 4.71 | 5.15 | 1.1 | 0.8 | 2.5Y 5/2 | 10YR 6/3 | Ehenge Poor - |
| Ehenge | NDOB_13 | Ondobe | 1.1 | 0.9 | 4.4 | 94.1 | 5.5 | 92.7 | 4.58 | 6.05 | 0.9 | NA | 10YR 6/3 | 10YR 4/3 | Ehenge Poor + |
| Ehenge | NDOB_19 | Ondobe | 1.5 | 1.1 | 2.9 | 95.5 | 5.3 | 93.2 | 4.87 | NA | 0.6 | 1 | 10YR 5/4 | 10YR 5/2 | Ehenge Poor + |
| Ehenge | OILYA_02 | Ondobe | 1.1 | 0.8 | 4.7 | 94.5 | 0 | 100 | 4.48 | 4.48 | 0.6 | 0.4 | 7.5YR 5/4 | 7.5YR 5/2 | Ehenge Poor - |
| Elondo | NDOB_20 | Ondobe | 1.9 | NA | 2.9 | 95.4 | 5.5 | 93.4 | 4.69 | 4.69 | 0.9 | NA | 7.5YR 5/4 | 5YR 5/2 | Elondo Poor + |
| Elondo | NDOB_08 | Ondobe | 1 | NA | 3.8 | 94.8 | NA | 97.1 | 5.17 | NA | 0.2 | 0.8 | 7.5YR 4/4 | 5YR 3/4 | Elondo Poor + |
| Elondo | NDOB_01 | Ondobe | 1.5 | 1.1 | 3.9 | 95 | 4.7 | 94.4 | 5.27 | 6.96 | 0.7 | 0.5 | 7.5YR 4/6 | 5YR 3/4 | Elondo Good 0 |
| Omufitu | OHNG_01 | Ondobe | 1.8 | 1.8 | 7.2 | 90.1 | 6.9 | 86.4 | 5.97 | NA | 0.2 | 0.2 | 7.5YR 4/6 | 7.5YR 4/3 | Omufitu Poor + |
| Omufitu | ETOPE_01 | Ondobe | 1.2 | 1.4 | 4.7 | 91.4 | 12 | 85.5 | 4.98 | NA | 0.1 | 0.1 | 10YR 3/2 | 10YR 3/2 | Omufitu Poor + |
| Omufitu | OMDI_03 | Omhedi | NA | NA | 6.5 | 94.4 | NA | NA | 6.49 | NA | 0.5 | NA | 10YR 6/3 | 7.5YR 5/4 | Omufitu Poor 0 |
| Omutunda | OMDI_02 | Omhedi | 4.4 | 2.3 | 9 | 87 | 22.6 | 73.9 | 5.97 | 6.52 | 0.5 | 0.7 | 10YR 3/2 | 10YR 3/2 | Omutunda Very good - |
| Omutunda | OMDI_01 | Omhedi | 3.2 | 2.9 | 28.5 | 63.3 | 29.8 | 65.3 | 7.69 | 7.69 | 1.4 | 1.5 | 10YR 4/1 | 10YR 4/1 | Omutunda Good + |
| Omutunda | HNDIB_02 | Ekolola | 1.3 | 1.4 | 3.1 | 95.4 | 5.5 | 93.4 | 4.69 | 4.69 | 0.1 | 0.1 | 10YR 4/3 | 7.5YR 4/3 | Omutunda Good + |
| Omutunda | NDOB_20 | Ondobe | 1.9 | NA | 3.8 | 94.8 | 2.9 | 97.1 | 5.09 | NA | 0.1 | NA | 7.5YR 4/3 | 7.5YR 4/4 | Omutunda Good - |
| Omutunda | NDOB_12 | Ondobe | 1.6 | 1.2 | 4.2 | 94.7 | NA | NA | 5.17 | NA | 0.1 | NA | 7.5YR 4/4 | 5YR 3/4 | Omutunda Good + |
| Omutunda | NDOB_03 | Ondobe | 2.6 | 1.9 | 6.6 | 91.6 | 13.1 | 85 | 5.14 | 7.15 | 0.5 | 0.8 | 7.5YR 4/1 | 2.5Y 4/1 | Omutunda Very good + |
| Omutunda | EFIDI_06 | Ondobe | 1.6 | 1.7 | 27.1 | 69.7 | 29.5 | 67.1 | 6.59 | 6.96 | 0.5 | 0.5 | 10YR 3/3 | 10YR 3/2 | Omutunda Good - |
| Omutunda | EFIDI_04 | Ondobe | 2 | 1.4 | 14 | 83.1 | 12.5 | 86 | 6.85 | 6.68 | 0.3 | 0.2 | 10YR 5/2 | 10YR 3/2 | Omutunda Good + |
| Omutunda | NDOB_14 | Ondobe | 1.4 | 1.2 | 19.2 | 78.6 | 17.8 | 80.1 | 7.48 | 7.67 | 0.2 | 0.5 | 10YR 3/3 | 10YR 4/1 | Omutunda Degraded + |
| Omutunda | NDOB_15 | Ondobe | 2.1 | 1.9 | 9.9 | 87.2 | 13.7 | 83 | 7.68 | 7.67 | 0.9 | 1 | 10YR 5/2 | 10YR 5/2 | Omutunda Degraded + |
| Omutunda | NDOB_16 | Ondobe | 2.9 | 1.9 | 8.1 | 89.8 | 10.7 | 86.4 | 7.36 | 6.4 | 0.9 | 0.4 | 10YR 4/2 | 10YR 5/2 | Omutunda Good 0 |
| Omutunda | NDOB_17 | Ondobe | 2.6 | NA | 6.1 | 91.7 | 11.8 | 86.5 | 7.7 | 7.19 | 0.6 | 0.5 | 10YR 4/2 | 10YR 4/2 | Omutunda Degraded + |
| Omutunda | OILYA_01 | Ondobe | 5 | 2.4 | 19.9 | 75.7 | 22.2 | 75 | 7.33 | 7.39 | 1.2 | 1.2 | 10YR 4/1 | 10YR 4/1 | Omutunda Very good - |
| Omutunda | OILYA_04 | Ondobe | 2.1 | 2.2 | 9.9 | 87.4 | 20.7 | 76.6 | 6.59 | 6.3 | 1.1 | 0.5 | 2.5Y 5/2 | 10YR 4/1 | Omutunda Good - |
| Omutunda | EKOL_01 | Ekolola | 1.7 | 2 | 7.2 | 90.5 | 10.4 | 86.6 | 4.74 | 5.16 | 0.5 | 0.2 | 10YR 5/4 | 10YR 4/3 | Omutunda Good 0 |
| Omutunda | HNDIB_01 | Ekolola | 2.9 | NA | 8 | 89.7 | 12.2 | 85.7 | 5.52 | 2 | 0.4 | NA | 10YR 4/2 | NA | Omutunda Degraded 0 |
| Omutunda | NGYO_01 | Ekolola | 1.6 | 1.6 | 4.3 | 93.9 | 10.2 | 86.6 | 5.01 | 5.05 | 0.2 | 0.2 | 10YR 5/2 | 10YR 4/2 | Omutunda Degraded - |




Table 6: Summary of the chemical and physical characteristics of topsoil (5-15 cm) and subsoil (25-35 cm)
layers of the studied *omutunda* soil profiles. CV= coefficient of variation.

| | | n | Min. | Median | Mean | Max. | CV |
|---|---|---|---|---|---|---|---|
| TOC (mg g⁻¹) | Top | 15 | 0.14 | 0.21 | 0.25 | 0.54 | 0.56 |
| | Sub | 11 | 0.12 | 0.19 | 0.2 | 0.29 | 0.25 |
| < 20 µm (%) | Top | 15 | 4.3 | 9 | 12 | 28 | 0.86 |
| | Sub | 15 | 63 | 87 | 85 | 94 | 0.1 |
| Sand (%) | Top | 15 | 10 | 13 | 17 | 30 | 0.52 |
| | Sub | 15 | 65 | 85 | 81 | 87 | 0.09 |
| pH (CaCl₂) | Top | 15 | 4.7 | 6.6 | 6.5 | 7.7 | 0.16 |
| | Sub | 15 | 5 | 6.8 | 6.7 | 7.7 | 0.13 |
| Moist colour value | Top | 15 | 0.2 | 0.5 | 0.64 | 1.4 | 0.77 |
| | Sub | 14 | 0.2 | 0.5 | 0.64 | 1.5 | 0.83 |

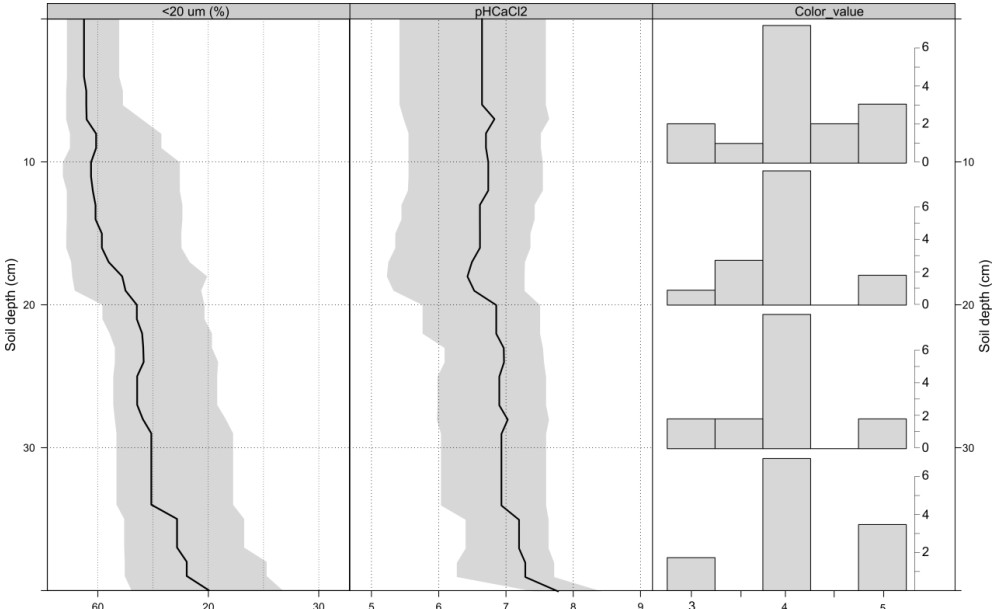

Figure 2: Results distribution of <20 µm fraction, pH(CaCl₂) and colour shade value of described *omutunda*. For
<20 µm fraction and pH (CaCl₂) variables, bold black lines represent the median value (surrounded by area
representing 25 and 75 percentiles). For colour shade values, frequency distribution is shown at 10, 20, 30 and
40 cm depth.
**3.4    Development of a soil quality evaluation toolbox**
**3.4.1    Importance of a soil quality evaluation toolbox**
We showed that KwSUs represent locations in the fields with specific soil characteristics and provide
information about their potential productivity. It notably includes soil hydraulic characteristics. Clearly, the
KwSU knowledge is land use orientated (e.g. suitability for pearl millet, workability), adapted to local conditions
(rainfall variability) and represents the local soil productivity potential. Farmers also include crop health, soil
consistence and colour shade to evaluate SQ of a specific location (Sect. 3.1). We also showed that each KwSU



includes a large variety of soils (especially *omutunda*) for which SQ for pearl millet production differs. To
estimate SQ, it is therefore important to standardise the assessment of the SQ at a specific location and time. This
would allow a comparison based on, for example, agricultural or climatic cycles or management techniques.
Technical soil characterisation (e.g. soil texture, colour) proved to be suitable to standardise SQ assessment in
other locations (Niemeijer and Mazzucato, 2003). We will therefore first show the meanings of the soil
characteristics used by the farmers to evaluate SQ and link these with soil technical analyses. Based on these
links, we will suggest ways to use this knowledge and to standardise the SQ assessment.

### 3.4.2 Important characteristics for field soil quality evaluation

Soils with a higher proportion of <20 μm particles are harder (Welch's $F_{(3, 55.3)} = 28.46$, $p$-value< 0.01; Figure
3), supported by more specific studies (Harper and Gilkes, 2004; Rawls and Pachepsky, 2002) and have larger
area of active surfaces, which play important role to fix SOC and nutrients (Feng et al., 2013). Through talking
about hardness, farmers indirectly refer to the proportion of fine soil particles (Osbahr and Allan, 2003). It
therefore indicates a major chemical property contributing to fertility. The proportion of <20 μm fraction content
in soils was increased through homestead shifting (clay-bricks remains) or mining riverbeds (Kreike, 2013).
Sand content (>63 μm) can be used to estimate the proportion of <20 μm fraction given the good correlation
between the proportion of these two classes ($p$-value< 0.01, $R^2 = 0.98$). We referred to it as the *potential chemical*
*fertility* because it requires appropriate fertilisation to fully achieve maximum fertility.

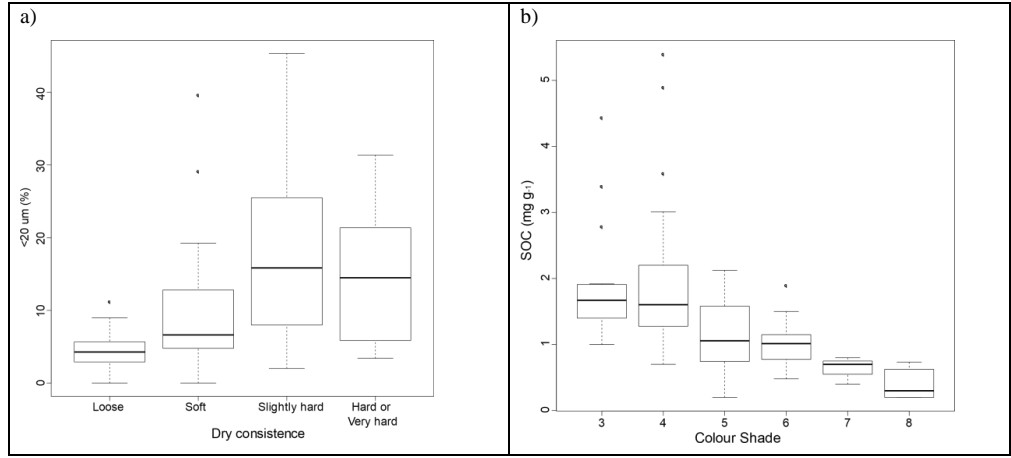

Figure 3: Boxplot showing the relation between a) fine particle (<20 μm) content and soil dry consistence and b)
soil organic content (SOC) and moist colour shade (Munsell colour value).
Soil colour shade is correlated with SOC of soils (Spearman's rank correlation rho $(-6.68, 108) = -0.54$, $p$-
value< 0.01; Figure 3). FFE point to the importance of "soil darkness" to estimate SQ. Therefore, the importance
of SOC for SQ is acknowledged. SOC is used as an index of SQ in many studies because of sensitivity to
management practices (Barrios and Trejo, 2003; Lima et al., 2011; Musinguzi et al., 2015; Osbahr and Allan,
2003). Sanchez et al. (2003) used the concept of SOC saturation (C-saturation) to evaluate the Soil Fertility
Capability and a C-saturation above 80 % indicated good soil conditions (Sanchez et al., 2003). For various





textural classes, SOC of undisturbed soils was calculated (Feng et al., 2013) and the colour shade value related to
it was estimated (Table 7)(Blume et al., 2011, p.51).
Table 7: Calculated soil organic carbon content (SOC) of 80%-C-saturated soil for various sand content (Feng et
al., 2013)and the estimated colour shade value (Blume et al., 2011, p.51).

| Sand content (%) | Saturated SOC (mg C g$^{-1}$ soil) | Optimal colour shade values |
|---|---|---|
| 80 | 6.8 | 3.5 |
| 85 | 5.5 | 3.5-4 |
| 90 | 4.2 | 3.5-4 |
| 95 | 3.0 | 4-4.5 |

**3.4.3  The soil quality evaluation toolbox**
Based on the link between FFE and soil technical properties, a toolbox for evaluating SQ based on indicators
adapted to Western Ohangwena region was developed. With this toolbox, SQ is assessed in two steps (Figure
4a): 1) Field participatory mapping of KwSUs; 2) technical SQ evaluation at specific locations using soil colour
shade and sand content.
As discussed, with KwSUs, farmers classify soils with comparable internal properties and suitability for pearl
millet production (Table 4). The distribution of KwSUs in the fields is known by most household members.
With participatory mapping, the farm can therefore be divided into KwSUs (*omutunda¸ ehenge*, *ehenene*, *elondo*
and *omufitu*), which represent internal soil properties.
Subsequently, soils are divided into three textural categories: <80 %, 80-90 %, and >90 % sand (Figure 4b)
representing textural limits discussed in various classifications (e.g. IUSS Working Group WRB, 2014). The
classes can be estimated in the field using texture-by-feel (Vos et al., 2016) or the Kruedener test adapted for
sandy soils (Fabry and Lutz, 1950; Nostitz, 1934). The three classes represent the evolution from good or
improved and to very poor chemical fertility potential, or degraded state for *omutunda*. *Elondo* are fine sandy
soils and coarse texture (>90%) indicates important degradation. Conversely, the proportions of sand are very
high in *ehenge* and *omufitu* (Table 5) and <90% sand indicates that major soil improvements had been
undertaken (e.g. former homestead location). Plant growth in *ehenene* is limited by the high soil pH and high
runoff intensity (Rigourd et al., 1999) and the soil texture is not relevant for SQ evaluation for this specific
KwSU.
Theoretical colour shade value of C-saturated soils vary from 3.5 for fine soils (<80 % sand) to 4.5 for very
coarse soils (>95 % sand; Table 7) (Blume et al., 2011, p.51). The three levels indicate fertilisation status.
*Positive* meaning sufficient organic inputs and *negative* meaning largely missing inputs. Munsell colour charts
are a standardised tool commonly used to evaluate bulk soil colours. The charts are relatively expensive, but
affordable for regional agricultural offices and are available for researchers from most soil science research
groups. Soil samples previously evaluated could also be used as comparison basis.
To make the evaluation closer to FFE, we suggest adapting the colour value scale for *ehenge* and *ehenene*
(optimal colour value +1) because these soils are lighter than the other KwSUs ("in *ehenge* the soil will look
white", KS, 60, Ondobe) and cannot reach low colour values.



a)

| Step 1: LSQI<br>Participatory mapping of KwSUs | **+** | Step 2: TSQI<br>Sand content & colour shade |
|---|---|---|

Semi-quantitative SQ evaluation

b)

| | LSQI | | TSQI | | | |
|---|---|---|---|---|---|---|
| **KwSU** | **Particularities** | **Sand content** | **Qualifier** | **Color shade value (moist)** | **Qualifier** | |
| **Omutunda** | Problematic during drought | >90 % | **Degraded** | 4 or less<br>4.5<br>5 or more | **+**<br>**0**<br>**-** | |
| | | 80-90 % | **Good** | 3.5 or less<br>4<br>4.5 or more | **+**<br>**0**<br>**-** | |
| | | <80 % | **Very good** | 3 or less<br>3.5<br>4 or more | **+**<br>**0**<br>**-** | |
| **Elondo** | | >90 % | **Degraded** | 4 or less<br>4.5<br>5 or more | **+**<br>**0**<br>**-** | |
| | | <90 % | **Good** | 3.5 or less<br>4<br>4.5 or more | **+**<br>**0**<br>**-** | |
| **Omufitu** | | >90 % | **Poor** | <4<br>4<br>>4 | **+**<br>**0**<br>**-** | |
| | | <90 % | **Improved** | | | |
| **Ehenene** | | >90 % | **Very poor** | 5 or less<br>5.5<br>6 or more | **+**<br>**0**<br>**-** | |
| | | <90 % | **Very poor** | 4.5 or less<br>5<br>5.5 or more | **+**<br>**0**<br>**-** | |
| **Ehenge** | Good during droughts | >90 % | **Poor** | 5 or less<br>5.5<br>6 or more | **+**<br>**0**<br>**-** | |
| | | <90 % | **Improved** | | | |

Figure 4: a) Schematic representation of the suggested SQ toolbox. It integrates Local Soil Quality Indicators
(LSQI) and Technical Soil Quality Indicators (TSQI) to create a semi-quantitative evaluation. b) Hierarchical SQ
evaluation. The evaluation starts with LSQI and classifies location into Oshikwanyama Soil Units (KwSU),
afterwards technical assessment is used to determine, chemical fertility potential (sand) and the soil organic
carbon (SOC) status (colour).
**3.4.4    Outcome of toolbox application**
The developed toolbox is a suggestion to evaluate SQ and to prioritise SQ-improvement practices. The resulting
SQ assessment gives a number of values, which bring more information about improvement potential than a
single value (Ditzler and Tugel, 2002). The various locations are therefore classified in a three levels system





(*KwSU*, *chemical fertility potential*, *SOC status*). KwSU represent internal soil properties that usually cannot be
modified in short term. Sand content indicates the *potential chemical fertility* of the soil, which can be improved
only with medium-term (decade) management practices (homestead relocation, erosion reduction). Colour shade
indicates the *SOC status* and can be modified in short-term, by fertilisation techniques (e.g. manuring,
conservation tillage).
The toolbox output provides three-value estimates that need to be interpreted based on local soil knowledge and
socio-economic context. For example, a soil can be characterised, by "ehenge poor+" (Table 5), which means
that: 1) The location undergoes waterlogging and is valuable during poor rainfall years (*ehenge*); 2) the *chemical*
*fertility potential* is low (*poor*); and 3) it is well enriched with organic materials (+). Investment to improve SQ
at this location could then focus on waterlogging risk reduction or clay enrichment, because strategies
concerning SOC are already adapted to the location and ameliorate SOC status would barely improve SQ and
productivity.
The test represents a way to estimate current soil status and it is therefore relevant to survey SQ in NCN. The
soils described during this study present a large diversity of SQ based on the developed SQ toolbox (Table 5).
Half the described *omutunda* (7/15) would need more organic inputs and five are considered degraded. These
results highlight the threat that exists for each location and indicate the measures to prioritise for SQ
improvements. Because of the lack of long term productivity data, it cannot be used to estimate the productivity
potential of a location. However, it would be relevant to guide, for example, the systematic collection of yield
data.
**4   Conclusion**
We developed a locally adapted method for SQ evaluation. Using the toolbox with farmers in NCN showed that
it is practical, affordable, precise and relatively easy to interpret. The suggested toolbox combines participatory
soil mapping with sand content and colour shades assessment. The toolbox fulfils the following conditions: (i)
practical and easy to use under field conditions; (ii) relatively precise and easy to interpret; (iii) relatively
economical; (iv) sufficiently sensitive to reflect the impact of soil use and management; (v) integrates physical,
chemical and biological characteristics and processes, and (vi) be useful for estimating soil properties or
functions that are difficult to measure.
The combination of farmers' and technical assessment cumulates advantages of both systems of knowledge,
specifically, the integrated long-term knowledge of the farmers (i.e. long-term productivity) and a short- (colour)
and medium term (sand fraction) SQ status assessment, sensitive to land management practices. The toolbox can
be used jointly by farmers and researchers from all fields of studies.
The toolbox represents a step towards better SQ evaluation in NCN. While it is adapted to a restricted area,
similar approaches can be used to develop SQ tools for areas where small-scale family agriculture represents
large proportion of land use. The results strongly support the use of FFE as entry point to SQ assessment at the
regional level, especially in semi-arid regions with high climatic variability and limited resources for SQ
assessment.



**5    Author contribution**
B. Prudat, L. Bloemertz and N.J. Kuhn designed the research. B. Prudat collected and interpreted the data. B.
Prudat prepared the manuscript with contributions from all co-authors.
**6    Competing interests**
The authors declare that they have no conflict of interest.
**7    Acknowledgements**
The research necessary for this paper was possible through the SNSF-DFG funded project: Communal land
reform in Namibia - Implications of Individualisation of land tenure. The authors gratefully thank all the
informants for sharing their knowledge and their reception into their home and the translators for their help
during the field work. The authors thank also the headmen and headwomen, constituency leaders, the Governor
of Ohangwena region and the director of the Ministry of Land and Resettlement for facilitating field work. We
thank the Polytechnic of Namibia for their collaboration, in particular A. Verlinden and D. Wyss for their advice
and help. We thank also our colleagues for laboratory work, advice and reviews.



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
