# Peer review of "Local soil quality assessment of north-central Namibia: integrating farmers' and technical knowledge"

_SOIL, 2017_

## Referee Comment (RC1) · A. Groengroeft (Referee) · 18 Jun 2017

General comments:

On page 3, the technical knowledge is highlighted because it facilitates international communication. Keeping this in mind, I wonder, why the authors do not try to present some information about the soil classification (reference groups and qualifier) according to FAO of the studied soils. The system is applied in Namibia and thus, from my perspective, this seems to be necessary.

Both variables in the SQ toolbox (sand content and color shade) are not independent

and are known by the local farmers in its indicative value. Although SOC is undoubtedly a very relevant variable for SQ, the direct link to color shade with one unit discriminating between the qualifier + and − is an over-interpretation of the possibilities of soil color interpretation. As given in figure 3, there is a significant overlap of SOC between neighboring color shade classes. The Munsell Soil Color Charts do not present colors (also not figure 3) for the broken classes as given in figure 4. Thus in the field very slight differences in color divide between the qualifier + and -, if the evaluator cannot decide, the qualifier becomes 0. Thus for me the combination of the variables fine particles and color is relevant, however, it is not promising to distinguish between 29 classes, as has been proposed in the toolbox by the authors.

The general problem of smallholder agriculture in the studied region is: i) Soils best suitable for cropping become scarce, thus expansion in the pristine woodlands will become increasingly restricted. ii) In the consequence, also those soils are cultivated, of which the farmers know their lower productivity. iii) the ongoing crop production is especially restricted by the lack of nutrient inputs, here N and P, and – off course – years with low rainfall. The future challenge is i) to concentrate crop production on the best suitable soils and ii) to improve nutrient inputs on these areas in an intensity, that yields are just water or management controlled and iii) to develop sustainable LU management techniques (e.g. conservation agriculture). This development needs help by the agriculture extension services. The mapping of the best suitable soils should be oriented to technical knowledge for its comparability, however should include farmers views. The general objective of the paper just moves to the right direction, the presented toolbox however needs improvement (reduction in units).

Please also note the supplement to this comment:
http://www.soil-discuss.net/soil-2017-10/soil-2017-10-RC1-supplement.pdf

**Supplement:**

Reviewer comments

"Local soil quality assessment of north-central Namibia: integrating farmers' and technical knowledge"

by Brice Prudat et al.

MS Version 3

Specific

| page | line | comments |
|---|---|---|
| 3 | table, line pH | I suggest, that doing numerous measurements on soil pH is cheaper by application of the sensor technique instead of the Hellige test kit |
| 4 | table | add a row with information, for which sort of soils (types, region) and land use the soil quality indicators were proposed |
| 4 | 10 | Verlinden and Dayot, 2005a |
| 6 | 31 | what is meant with color shade ? The standardized MUNSELL soil color charts are composed by the variables hue, value and chroma. Is shade identical with value ? Please explain. |
| 7 | 12 | "two sample rings": defined volume? calculation of bulk density possible ? please explain or reformulate |
| 9 | table | hardness is an often mentioned indicator for soil quality. I suggest, that the hardness is related to the condition of the soil in the (almost) dry state, perhaps for that time of the year, when ploughing is done. Please add some explanations on the local farmers intention. |
| 10 | 21 | values of pH (8.4 to 10.1) are not existing in Table 5 ! |
| 11 | table 5 | in row <20 μm – sub data of sand are give and in row sand – sub data of < 20 μm are given.  Check all data and compare with data in respective chapters.

add row with WRB classification |
| 12 | table 6 | same mistakes as for table 5 !! |
| 12 | 6 | acc. to tab. 6 the coefficient of variation is large for TOC, moist color value, < 20 μm fraction but not for pH (in both depth intervals CV < 0.2) |
| 13 | figure 2 | this graph pretends a precise depth distribution which was not analysed. Additionally this graph is redundant, please delete. |
| 13 | 8 | fragipan: delete term because of its vague definition |
| 14 | 8 | "large variety of soils" → large variety of soil properties |
| 14 | 9 | "standardize the assessment of the SQ at a specific location and time". Why time ? Soil quality assessments always results in a potential for intended land use. Different climatic conditions may be included in the potential. Thus the results are irrespective of time, however may be altered by changes in soil properties due to land use.
What is meant with location in this context: Three villages were studied, should the SQ assessment by different for each village? |
| 14 | 16 | harder in dry conditions (?) |
| 14 | 21 | this increase in < 20 μm can only be marginal |
| 15 | 12 | Data presented by Blume eta al 2011 cannot be transferred to Namibian soils. |

| 16 | 8 | "indicates important degradation". Relevant forms of degradation (acidification, salinization, decline in nutrients, compaction) do not include the shift to more coarse particles. Thus, if farmers classify soils as Elondo and field tests show, that sand content is > 90 %, this means that farmers needed to shift to less suitable soils. |
|----|---|----|
| 16 | 9 | "major soil improvements": see above |
|    |   |    |

---

## Referee Comment (RC2) · Anonymous Referee #2 · 17 Oct 2017

This is a very interesting and valuable work which deals with local soil assessment in Namibia by combining ethnopedology (eg. local farmers knowledge) and soil science. A new soil quality toolbox is provided to evaluate soil quality in a fast, cheap and easy way, which can be used by local farmers and researchers. From my point of view, it is an important paper, which could be improved significantly by using an international soil classification (WRB) and description of the different KwSus, making it accessible for a wider audience and allow for international comparison and land management studies in other areas comprising comparable environmental conditions.

This manuscript is written clearly and the data is well documented, however photographic documentation of soil profiles (if available) and profile descriptions seem appropriate making it more attractive and better accessible to the readers.

The methodology seems generally adequate. However, I don't fully agree with the argument (page 7, line 23) against the measurement of the Cations exchange capacity. As high contents of carbonates and salts are expected it could be important to know which kind of salts are present to be able to adapt land management.

Also, it is not clear how soil fertility/chemical fertility, used in results and discussion, is defined in this study: Is it the potential of the soil to provide nutrients coming from natural sources or artificial with fertilizers? Or the plant available nutrients? In the results and discussion, the authors refer to chemical fertility, I would suggest replacing chemical fertility with soil fertility, as chemical fertility includes available nutrient contents, which were not measured. In section 3.2, page 10: The authors suggest a high chemical fertility and chemical exchange capacity for the omutunda units. This is misleading since it gives the reader the feeling that this soil is highly fertile. It should be made clear that this is relatively seen.

Fig. 1 needs a reference of the satellite image and hydrology data. A little box indicating the section of the study area in the map of Namibia would be useful. Fig. 2 needs some clarification as it seems that pH and <20$\mu$m content were measured in high resolution and vary in depth.

---

## Referee Comment (RC3) · J. van Leeuwen (Referee) · 19 Oct 2017

The paper presents interesting work, that is potentially valuable and fitting the scope and quality of SOIL. It combines local farmers knowledge and soil scientific practices in a toolbox for soil quality evaluation and is thereby applicable in many parts of the world where resources are scarce (but even in Europe this approach holds its value). There are however a few points that can be improved. Including soil descriptions using WRB soil classification increases the relevance to the broader public. The presented toolbox seems useful, but detecting very slight colour differences in the field will not be easy.

As in many tropical agricultural soils, fertility in terms of N and P availability will be a

severe limiting factor in this area (besides water limitation). It is however not taken into account in the soil quality evaluation. I realize that it may not be a property that can readily be measured by farmers, but it should at least be discussed as an important limiting factor.

A few specific comments:

Table 1:

- depth of topsoil can better be changed to soil depth or rooting depth, as depth of topsoil is defined by the user, not so much a soil property.

- Infiltration rate, or capacity?

P3L7: Soil diversity: misleading term, soil variability is more apt.

P3L14: How do you define the process of agricultural evolution?

P6L8: unclear why some farmers are visited more than once, while others are not.

P7L21: Further on only pHCaCl is shown/mentioned, so why also include pHH2O here? Better remove it if you don't show further results.

P10L2: chemical fertility is still low compared to many other soils. Differences are relative between local soils, which should be emphasized. Also the term chemical fertility may be a bit misleading; soil fertility may be better in this context.

Figure 2 doesn't seem to be very relevant for the story, not very comparable to the other data shown (more detailed). So I would suggest to remove it. Also values on x-axes of first and third pane are hard to understand (not in line with table above).

P14L17-26: sentences are hard to understand. Wording can be improved/clarified. Maybe replace evolution by transition? Improvement in this context is are to follow, it seems to imply that improvement has taken place over time, but without reference in the past? What were the conditions before the improvement?

Technical/textual points:

P3L20: have been developed and discussed, and yielded ...

P4L1: farmers and technical assessments,

P6L2: remove space after Sandveld

P6L15: insert second closing bracket after 2005.

P7L26: replace that by when

P10L32: various entities ...

P13L5: meaning

P13L11: play an important role in fixing

---

## Author Comment (AC1) · 16 Nov 2017

We are very excited to have been given the opportunity to revise our manuscript and are very thankful for the number and pertinence of the comments.

Herein, we explain how we answer those comments and suggest how we would revise the paper.

Reviewer's comment: On page 3, the technical knowledge is highlighted because it facilitates international communication. Keeping this in mind, I wonder, why the authors do not try to present some information about the soil classification (reference groups

and qualifier) according to FAO of the studied soils. The system is applied in Namibia and thus, from my perspective, this seems to be necessary.

Author's answer and suggestions: We agree with this concern and we would add few soil descriptions, including pictures and FAO names. A short chapter following 3.2 "Technical analysis of farmers' field experiences" will be added to explain the major results regarding the WRB (reference soil groups, qualifiers). Add in Chapter 3.5.1 "Importance of a soil quality evaluation toolbox": "Soil classification based on the FAO is used by Namibian institutions and is used to draw the Namibian soil map. Therefore, it could be appropriate to use for international and scientific communication. However, this classification system does not bring additional information that would benefit this paper and was therefore not discussed. FAO classification is orientated towards representing "primary pedogenetic process[es]" and does not aim at detecting soil differences at micro-scale, neither spatial nor temporal" (IUSS Working Group WRB, 2014). Therefore, the use of this classification is not relevant to highlight SQ differences at small-scale. Moreover, the classification of the described soils in the WRB is poorly informative given the low prevalence of diagnostic properties and horizons leading to poorly informative nomenclature."

Reviewer's comment: Both variables in the SQ toolbox (sand content and colour shade) are not independent and are known by the local farmers in its indicative value.

Author's answer and suggestions: It is true that sand content and colour shade are not independent variables. Despite the dependency between these variables, we can use both to evaluate SQ because meaningful variability remains. The indicative value of these properties is known by the farmers. As explained in the introduction, farmers' knowledge is valuable but lack of standardisation, which can be brought by technical assessment.

Reviewer's comment: Although SOC is undoubtedly a very relevant variable for SQ, the direct link to colour shade with one unit discriminating between the qualifier + and – is

an over-interpretation of the possibilities of soil colour interpretation. As given in figure 3, there is a significant overlap of SOC between neighbouring colour shade classes. Thus, in the field very slight differences in colour divide between the qualifier + and –, if the evaluator cannot decide, the qualifier becomes 0.

Author's answer and suggestions: It must be emphasized that this toolbox is a suggestion that would require further developments. In §3.4.4 "Outcome of toolbox application": "The developed toolbox is and remains a suggestion for evaluating SQ and for prioritising SQ-improvement practices." Considering the comment and the low accuracy of Munsell colour evaluation, we would modify the colour shade classes defined in the toolbox (Fig 4b) to include more soil colour shade values in the 0 class. This would result in more soils classified into this class; 1) avoiding an overinterpretation of SQ evaluation and 2) corresponding better to the farmers SQ evaluation. The values are then adapted to avoid over-interpretation of field data collected.

Reviewer's comment: The Munsell Soil Color Charts do not present colours (also not figure 3) for the broken classes as given in figure 4.

Author's answer and suggestions: The broken classes suggested in Fig. 4 are defined based on theoretically-calculated optimal SOC and colour shade values.

Reviewer's comment: The combination of the variables fine particles and colour is relevant, however, it is not promising to distinguish between 29 classes, as has been proposed in the toolbox by the authors.

Author's answer and suggestions: It leads indeed to 29 classes possible. However, this classification is constructed as a combination of 5 KwSUs, 4 texture classes and 3 colour classes. Each level has a specific meaning and can be evaluated without the other (e.g. "-" for colour value indicates a need for organic fertilisers, no regards with KwSU or texture. In §3.4.4, we would add a comment considering this high number of classes and emphasize that all classification levels can be used separately.

Reviewer's comment: The general problem of smallholder agriculture in the studied region is: i) Soils best suitable for cropping become scarce, thus expansion in the pristine woodlands will become increasingly restricted. ii) In the consequence, also those soils are cultivated, of which the farmers know their lower productivity. iii) the ongoing crop production is especially restricted by the lack of nutrient inputs, here N and P, and – off course – years with low rainfall.

Author's answer and suggestions: Known to be limiting nutrients in most agricultural land, particularly in sub-Saharan Africa, nitrogen and phosphor availability are most likely significant for plant growth. However, given the high connection between these nutrients and the soil short-term fertilisation, we decided to not include these analysis in our analysis as our work aim to understand and follow longer-term soil fertility discussion. In §2.3.2 Laboratory analyses: add "Known to be limiting nutrients in most agricultural land and in sub-Saharan Africa, nitrogen and phosphor availability are most likely significant for plant growth. However, given the high connection between these nutrients and the soil short-term fertilisation, we decided to not include these analysis in our analysis as our work aim to understand and follow longer-term soil fertility discussion."

Reviewer's comment: The future challenge is i) to concentrate crop production on the best suitable soils and ii) to improve nutrient inputs on these areas in an intensity, that yields are just water or management controlled and iii) to develop sustainable LU management techniques (e.g. conservation agriculture). This development needs help by the agriculture extension services.

Author's answer and suggestions: This issue would need to be discussed. However, it does not relate directly with the objectives of the current paper, which do not aim at suggesting management techniques to improve SQ. The aim is to suggest a SQ toolbox that helps to evaluate the conditions of a soil, regarding its potential.

Reviewer's comment: The mapping of the best suitable soils should be oriented to

technical knowledge for its comparability, however should include farmers views. The general objective of the paper just moves to the right direction, the presented toolbox however needs improvement (reduction in units).

Author's answer and suggestions: Our objective is not to create a map or tools to map, it is to enable the farmers to optimise their SQ evaluation. However, for mapping purposes, each criterion used can be mapped separately, which presents the advantage to evaluate the various issues separately (organic matter availability, erosion, soil types).

Reviewer's comment: P3, Table 1, L pH: I suggest, that doing numerous measurements on soil pH is cheaper by application of the sensor technique instead of the Hellige test kit

Author's answer and suggestions: We accept this proposition and add this method in the Table 1.

Reviewer's comment: P4, Table 2: add a row with information, for which sort of soils (types, region) and land use the soil quality indicators were proposed

Author's answer and suggestions: Very relevant, we decided to ignore this information to facilitate the reading of the Table. We would then modify Table 2 by adding this information.

Reviewer's comment: P6 L31: what is meant with colour shade? The standardized MUNSELL soil colour charts are composed by the variables hue, value and chroma. Is shade identical with value? Please explain.

Author's answer and suggestions: Colour shade in common language would refer to "colour value" in the Munsell colour system. We use both to separate the perception of shade (dark or light) against the numeric evaluation of darkness (colour value).

Reviewer's comment: P7 L12: "two sample rings": defined volume? calculation of bulk density possible? please explain or reformulate

**SOILD**
[Figure]

Author's answer and suggestions: "Two 100 cm3-sampling rings were collected from each described horizon. Dried-samples were weighted to calculate bulk density, sieved (2 mm) and used for further analysis."

Reviewer's comment: P9 Table 4: hardness is an often-mentioned indicator for soil quality. I suggest, that the hardness is related to the condition of the soil in the (almost) dry state, perhaps for that time of the year, when ploughing is done. Please add some explanations on the local farmers intention.

Author's answer and suggestions: The consistence, or the concept of hardness, is in relation to the hardness of dry soil, which impacts importantly the difficulty of ploughing. Add a clarification of the meaning of hardness at the beginning of the chapter 3 "Results and Discussion": The consistence, or the concept of hardness, is in relation to the hardness of dry soil, which impacts importantly the difficulty of ploughing.

Reviewer's comment: P10 L21: values of pH (8.4 to 10.1) are not existing in Table 5!

Author's answer and suggestions: These pH values are pH in water, while in Table 5 the values are pH in CaCl2. We would clarify this difference in the text.

Reviewer's comment: P11 Table 5 and P12 Table 6: Row <20 $\mu$m sub data of sand are given and in row sand – sub data of < 20 $\mu$m are given. Check all data and compare with data in respective chapters.

Author's answer and suggestions: Some errors are in the Table 5 and 6. We will modify the Table 5 and 6 to clarify the particle size content.

Reviewer's comment: P11 Table 5: Add row with WRB classification

Author's answer and suggestions: We don't think that WRB names for all soils would help the discussion because of the presence of very limited number of diagnostic horizon, properties or materials, which leads to very limiting meaning of WRB names. We would add few soil descriptions, including pictures and FAO names. A short chapter following 3.2 "Technical analysis of farmers' field experiences" will be added to explain the

most important characteristics regarding the WRB (reference soil groups, qualifiers).

Reviewer's comment: P12 L6: acc. to tab. 6 the coefficient of variation is large for TOC, moist color value, < 20 $\mu$m fraction but not for pH (in both depth intervals CV < 0.2).

Author's answer and suggestions: This error comes from a calculation mistake. To calculate CV of pH, we should not use the pH values but the H+ concentrations. Using these values, we find a CV of 1.2

Reviewer's comment: P13 Fig2: This graph pretends a precise depth distribution which was not analysed. Additionally, this graph is redundant, please delete.

Author's answer and suggestions: This remark is relevant, and we will delete this table.

Reviewer's comment: P13 L8: fragipan: delete term because of its vague definition

Author's answer and suggestions: Fragipan is the most accurate concept that describe this soil layer. We would add the definition of fragipan suggested by Soil Survey Staff (1999): "Fragipans are compacted layers that result from various processes (freezing-thawing, pressure, swelling-drying), excluding ploughing pan."

Reviewer's comment: P14 L8: "large variety of soils" -> large variety of soil properties

Author's answer and suggestions: Accept the change to "soil properties".

Reviewer's comment: P14 L9: "standardize the assessment of the SQ at a specific location and time". Why time? Soil quality assessments always results in a potential for intended land use. Different climatic conditions may be included in the potential. Thus, the results are irrespective of time, however may be altered by changes in soil properties due to land use.

Author's answer and suggestions: SQ represents the potential regarding various climatic conditions, but is also the consequence of various soil degradation or improvements techniques. Therefore, the notion of time is important. The SQ of a specific site

can change through agricultural activities.

Reviewer's comment: P14 L9: What is meant with location in this context: Three villages were studied, should the SQ assessment by different for each village?

Author's answer and suggestions: Not in this context. The assessment would be done following the same method in all villages. What we mean is that it is important to evaluate SQ at various location (points) and to compare the results between these locations. The comparison is important to evaluate the potential that can be reached in specific regions (villages). There is no need to explain that a soil in Ekolola (woodland) is bad and in Omhedi (Oshana) it is good. It is more useful to differentiate between various location in a same village, to evaluate the potentials.

Reviewer's comment: P14 L16: harder in dry conditions (?)

Author's answer and suggestions: Add "in dry condition" after harder.

Reviewer's comment: P14 L21: This increase in <20 $\mu$m can only be marginal.

Author's answer and suggestions: The increase in fine particle content can be significant by mining riverbeds for example, following researches from Kreike (2013), as already explained in the manuscript (P13 L14).

Reviewer's comment: P15 L12: Data presented by Blume et al 2011 cannot be transferred to Namibian soils.

Author's answer and suggestions: It is a relevant comment given the origin of the soils used in Blume 2011, difficult to compare to the Namibian context. However, we did not find similar relations adapted to tropical soils. Moreover, the results indicate a relatively well-balanced repartition of SOC status in our soils, which therefore helps to analyse the SQ status of a soil in relation to other soils of the same region.

Reviewer's comment: P16 L8: "indicates important degradation". Relevant forms of degradation (acidification, salinization, decline in nutrients, compaction) do not include

the shift to more coarse particles.

Author's answer and suggestions: Processes that can remove fine particles from the top soil are 1) eluviation related to dispersive salts; 2) overland flow erosion, 3) wind erosion. Add suggestions of processes leading to soil texture coarsening.

Reviewer's comment: P16 L8: Thus, if farmers classify soils as Elondo and field tests show, that sand content is > 90 %, this means. that farmers needed to shift to less suitable soils.

Author's answer and suggestions: This was observed with people who moved to the eastern areas (Woodlands), where they described omutunda that was not as fertile as the omutunda found in the western areas (Oshana). However, farmers who moved to or extended their fields to less fertile land will not necessarily classify this new land as a fertile KwSU. A sentence summarising this issue would be added in §"3.4.4 Outcome of toolbox application".

Reviewer's comment: P16 L9: major soil improvements": see above.

Author's answer and suggestions: The increase in fine particle content can be significant, following researches from Kreike (2013) (P13 L14).

We want to extend our appreciation for taking the time and effort necessary to provide such insightful guidance.

We hope that the answers and the suggested revisions improve the paper.

---

## Author Comment (AC2) · 16 Nov 2017

We are very excited to have been given the opportunity to revise our manuscript. We carefully considered your comments. Herein, we explain how we revised the paper based on those comments and recommendations.

We hope that these revisions improve the paper following your suggestions.

General comments

Reviewer's comment: From my point of view, it is an important paper, which could be improved significantly by using an international soil classification (WRB) and description of the different KwSUs, making it accessible for a wider audience and allow for international comparison and land management studies in other areas comprising comparable environmental conditions.

Author's answer and suggestions: We agree with this concern and we would add few soil descriptions, including pictures and FAO names. A short chapter following 3.2 "Technical analysis of farmers' field experiences" will be added to explain the major trends regarding the WRB (reference soil groups, qualifiers). Add in Chapter 3.5.1 "Importance of a soil quality evaluation toolbox": "Soil classification based on the FAO is used by Namibian institutions and is used to draw the Namibian soil map. Therefore, it could be appropriate to use for international and scientific communication. However, this classification system does not bring additional information that would benefit this paper and was therefore not discussed. FAO classification is orientated towards representing "primary pedogenetic process[es]" and does not aim at detecting soil differences at micro-scale, neither spatial nor temporal" (IUSS Working Group WRB, 2014). Therefore, the use of this classification is not relevant to highlight SQ differences at small-scale. Moreover, the classification of the described soils in the WRB is poorly informative given the low prevalence of diagnostic properties and horizons leading to poorly informative nomenclature."

Reviewer's comment: Photographic documentation of soil profiles (if available) and profile descriptions seem appropriate making it more attractive and better accessible to the readers.

Author's answer and suggestions: Soil descriptions and pictures would be added (See above)

Reviewer's comment: P7 L23: I don't fully agree with the argument against the measurement of the Cations exchange capacity.

Author's answer and suggestions: Change this section to clarify the decision (§Methods): "Cation exchange capacity and base saturation [...] were not measured in this

study because the presence of calcium carbonates and soluble salt strongly influences the measurements, which makes results very difficult to use for comparison, especially considering the low expected values due to low cation exchanging materials (mostly clay and organic matter)."

Reviewer's comment: §2.3.2 Laboratory analyses: As high contents of carbonates and salts are expected it could be important to know which kind of salts are present to be able to adapt land management.

Author's answer and suggestions: We agree with this comment and suggest adding some information concerning the type of salts expected in the area. Add this information in §2.3.2: "the presence of calcium carbonates (secondary precipitations observed in various soil profiles) and soluble salt (high EC in ehenene, mostly NaCl)."

Reviewer's comment: It is not clear how soil fertility/chemical fertility, used in results and discussion, is defined in this study: Is it the potential of the soil to provide nutrients coming from natural sources or artificial with fertilizers? Or the plant available nutrients?

Author's answer and suggestions: Make clear what chemical refers to... §3.2 Technical analysis of farmers' field experience: "All these characteristics suggest the higher potential of omutunda to provide nutrients, coming from any sources, compared to the other KwSUs. This capacity is hereafter called chemical fertility."

Reviewer's comment: In the results and discussion, the authors refer to chemical fertility, I would suggest replacing chemical fertility with soil fertility, as chemical fertility includes available nutrient contents, which were not measured.

Author's answer and suggestions: We always used "chemical fertility potential" to clearly indicate that it is not the actual chemical fertility (related to nutrient content) but an indicator for the potential that the soil could reach if sufficiently fertilised. We think that replacing "chemical fertility" by "soil fertility" will add confusion to the reader.

[Figure]

Reviewer's comment: §3.2, page 10: The authors suggest a high chemical fertility and chemical exchange capacity for the omutunda units. This is misleading since it gives the reader the feeling that this soil is highly fertile. It should be made clear that this is relatively seen.

Author's answer and suggestions: Changes in §3.2, P10 L1: "All these characteristics suggest the higher potential of omutunda to provide nutrients, coming from any sources, compared to the other KwSUs."

Reviewer's comment: Fig. 1 needs a reference of the satellite image and hydrology data.

Author's answer and suggestions: Add origin of the satellite images and hydrology data.

Reviewer's comment: Fig. 1: A little box indicating the section of the study area in the map of Namibia would be useful.

Author's answer and suggestions: Add the suggested box.

Reviewer's comment: Fig. 2 needs some clarification as it seems that pH and <20_m content was measured in high resolution and vary in depth.

Author's answer and suggestions: We would remove this figure, given the different depth resolution illustrated compared to the rest of the data used.

Again, we appreciate all your insightful comments and are thankful to you for taking the time and energy to help us improve the paper. We hope that the answers and the suggested revisions improve the paper.

---

## Author Comment (AC3) · 16 Nov 2017

We are very excited to have been given the opportunity to revise our manuscript. We carefully considered your comments. Herein, we explain how we revised the paper based on those comments and recommendations.

We hope that these revisions improve the paper following your suggestions.

General comments

Reviewer's comment: Including soil descriptions using WRB soil classification increases the relevance to the broader public.

Author's answer and suggestions: We agree with this concern and we would add few soil descriptions, including pictures and FAO names. A short chapter following 3.2 "Technical analysis of farmers' field experiences" will be added to explain the major trends regarding the WRB (reference soil groups, qualifiers). Add in Chapter 3.5.1 "Importance of a soil quality evaluation toolbox": "Soil classification based on the FAO is used by Namibian institutions and is used to draw the Namibian soil map. Therefore, it could be appropriate to use for international and scientific communication. However, this classification system does not bring additional information that would benefit this paper and was therefore not discussed. FAO classification is orientated towards representing "primary pedogenetic process[es]" and does not aim at detecting soil differences at micro-scale, neither spatial nor temporal" (IUSS Working Group WRB, 2014). Therefore, the use of this classification is not relevant to highlight SQ differences at small-scale. Moreover, the classification of the described soils in the WRB is poorly informative given the low prevalence of diagnostic properties and horizons leading to poorly informative nomenclature."

Reviewer's comment: The presented toolbox seems useful, but detecting very slight colour differences in the field will not be easy.

Author's answer and suggestions: It must be emphasized that this toolbox is a suggestion that would require further developments. Considering the comment and the low accuracy of Munsell colour evaluation, we would modify the colour shade classes defined in the toolbox to include more soil colour shade values in the 0 class. This would result in more soils classified into the "0"-class; 1) avoiding an overinterpretation of changes to be undertaken; and 2) corresponding better to the farmers SQ evaluation. The values are then adapted to avoid over-interpretation of field data collected. In §3.4.4 "Outcome of toolbox application": "The developed toolbox is and remains a suggestion for evaluating SQ and for prioritising SQ-improvement practices." Moreover, we would change the SQ evaluation classes of Colour shade (Fig 4b) to include more values in the 0-class.

Reviewer's comment: As in many tropical agricultural soils, fertility in terms of N and P availability will be a severe limiting factor in this area (besides water limitation). It is however not taken into account in the soil quality evaluation. I realize that it may not be a property that can readily be measured by farmers, but it should at least be discussed as an important limiting factor.

Author's answer and suggestions: Known to be limiting nutrients in most agricultural land, particularly in sub-Saharan Africa, N and P availability are most likely significant for plant growth. However, given the high connection between these nutrients and the soil short-term fertilisation, we decided to not include these analysis in our evaluation as our work aim to understand and follow longer-term soil fertility discussion. We would however add a comment in §2.3.2 Laboratory analyses, to explain why we did not measure these nutrients.

Specific Comments

Reviewer's comment: Table 1: depth of topsoil can better be changed to soil depth or rooting depth, as depth of topsoil is defined by the user, not so much a soil property.

Author's answer and suggestions: These properties have been copied from Wienhold et al. (2004), as suggested in the figure caption. We would therefore not change it.

Reviewer's comment: Table 1: Infiltration rate, or capacity?

Author's answer and suggestions: Water infiltration rate, to precise in Table 1.

Reviewer's comment: P3L7: Soil diversity: misleading term, soil variability is more apt.

Author's answer and suggestions: Changed following suggestion.

Reviewer's comment: P3L14: How do you define the process of agricultural evolution?

Author's answer and suggestions: Changed into "evolution of agricultural practices" (P3L13).

Reviewer's comment: P6L8: unclear why some farmers are visited more than once, while others are not.

Author's answer and suggestions: The farmers who showed a broad soil and agricultural knowledge during the first interview and open to discussion were visited several times.

Reviewer's comment: P7L21: Further on only pHCaCl is shown/mentioned, so why also include pHH2O here? Better remove it if you don't show further results.

Author's answer and suggestions: pHH2O removed from Methods.

Reviewer's comment: P10L2: chemical fertility is still low compared to many other soils. Differences are relative between local soils, which should be emphasized.

Author's answer and suggestions: Changes in §3.2, P10 L1: "All these characteristics suggest the higher potential of omutunda to provide nutrients, coming from any sources, compared to the other KwSUs."

Reviewer's comment: Also the term chemical fertility may be a bit misleading; soil fertility may be better in this context.

Author's answer and suggestions: We always used "chemical fertility potential" to clearly indicate that it is not the actual chemical fertility (related to nutrient content) but an indicator for the potential that the soil could reach if sufficiently fertilised. We think that replacing "chemical fertility" by "soil fertility" will add confusion to the reader.

Reviewer's comment: Fig 2 doesn't seem to be very relevant for the story, not very comparable to the other data shown (more detailed). So I would suggest to remove it. Also values on x-axes of first and third pane are hard to understand (not in line with table above).

Author's answer and suggestions: We would remove this figure, given the different depth resolution illustrated compared to the rest of the data used.

Reviewer's comment: P14L17-26: sentences are hard to understand. Wording can be improved/clarified.

Author's answer and suggestions: Change wording for better clarity. For example: "The three classes represent the transition from "good" (or "improved") to "very poor" (or "degraded") chemical fertility potential. Most elondo are fine sandy soils, in which coarse texture (>90%) would indicate ongoing or past degradation because elondo is described as a fertile soil.

Reviewer's comment: Maybe replace "evolution" by "transition"?

Author's answer and suggestions: Accept the suggestion.

Reviewer's comment: Improvement in this context is are to follow, it seems to imply that improvement has taken place over time, but without reference in the past? What were the conditions before the improvement?

Author's answer and suggestions: There is a lack of data to support the assumption of soil degradation or improvement. However, these processes were perceived and explained by some farmers during the interviews. Add this explanation after P15 L9.

Technical/textual points:

Reviewer's comment: P3L20: "have been developed and discussed, and yielded" ...

Author's answer and suggestions: Suggested correction accepted.

Reviewer's comment: P4L1: "farmers and technical assessments"

Author's answer and suggestions: We suggest using "between technical and farmers assessment".

Reviewer's comment: P6L2: remove space after "Sandveld"

Author's answer and suggestions: Suggested correction accepted.

Reviewer's comment: P6L15: insert second closing bracket after 2005.

Author's answer and suggestions: Suggested correction accepted.

Reviewer's comment: P7L26: replace "that" by "when"

Author's answer and suggestions: Suggested correction accepted.

Reviewer's comment: P10L32: "various entities" ...

Author's answer and suggestions: Suggested correction accepted.

Reviewer's comment: P13L5: "meaning"

Author's answer and suggestions: Suggested correction accepted.

Reviewer's comment: P13L11: "play an important role in fixing"

Author's answer and suggestions: Suggested correction accepted.

Again, we appreciate all your insightful comments and are thankful to you for taking the time and energy to help us improve the paper. We hope that the answers and the suggested revisions improve the paper.

---

## Author Response (AR1)

**Answer to reviewer 1**

| Position in manuscript | Reviewer's comments | Authors' answer | Changes on new manuscript |
|---|---|---|---|
| page 3 | the technical knowledge is highlighted because it facilitates international communication. Keeping this in mind, I wonder, why the authors do not try to present some information about the soil classification (reference groups and qualifier) according to FAO of the studied soils. The system is applied in Namibia and thus, from my perspective, this seems to be necessary. | We agree with this concern. However, given the lack of diagnostic properties, the WRB is poorly informative in this context. | Page 20 Table A1 (Appendix A): Add three soil descriptions, including pictures and WRB; Page 10, L 8: Add a short chapter following "International classification". Page 13 Table 6: Added table, in which all soil profiles are classified using the WRB. |
| General | Both variables in the SQ toolbox (sand content and color shade) are not independent and are known by the local farmers in its indicative value. | Despite the dependency between these variables, we can use both to evaluate SQ because meaningful variability remains. The indicative value of these properties is known by the farmers. As explained in the introduction, farmers' knowledge is valuable but lack of standardisation, which can be brought by technical assessment. | |
| General | Although SOC is undoubtedly a very relevant variable for SQ, the direct link to color shade with one unit discriminating between the qualifier + and – is an over-interpretation of the possibilities of soil color interpretation. As given in figure 3, there is a significant overlap of SOC between neighboring color shade classes. Thus, in the field very slight differences in color divide between the qualifier + and –, if the evaluator cannot decide, the qualifier becomes 0. | It must be emphasized that this toolbox is a suggestion that would require further developments. | Page 16 L30. |

| Position in manuscript | Reviewer's comments | Authors' answer | Changes on new manuscript |
|---|---|---|---|
| | | Considering the comment and the low accuracy of Munsell colour evaluation, we modified the colour shade classes defined in the toolbox in order to include more soil colour values in the neutral class (0). This result in more soils classified into this class, 1) avoiding an over-interpretation of changes to be undertaken and 2) corresponding better to the farmers SQ evaluation. The values are then adapted to avoid over-interpretation of field data collected. | Page 17 Table 10: Change the SQ evaluation classes of Colour shade. |
| General | The Munsell Soil Color Charts do not present colors (also not figure 3) for the broken classes as given in figure 4. | The broken classes suggested in figure 10 are defined based on theoretically-calculated optimal colour shade values. | |
| General | The combination of the variables fine particles and color is relevant, however, it is not promising to distinguish between 29 classes, as has been proposed in the toolbox by the authors. | The toolbox leads to 29 classes possibilities. However, this classification is constructed as a combination of 5 KwSUs, 4 texture classes and 3 colour classes. Each level has a specific meaning and can be evaluated without the other (e.g. "-" for colour value indicates a need for organic fertilisers, no regards with KwSU or texture. | Page 18 L. 3-17: emhasize that all classification levels can be used separately. |

| Position in manuscript | Reviewer's comments | Authors' answer | Changes on new manuscript |
|---|---|---|---|
| General | The general problem of smallholder agriculture in the studied region is: i) Soils best suitable for cropping become scarce, thus expansion in the pristine woodlands will become increasingly restricted. ii) In the consequence, also those soils are cultivated, of which the farmers know their lower productivity. iii) the ongoing crop production is especially restricted by the lack of nutrient inputs, here N and P, and – off course – years with low rainfall. | Known to be limiting nutrients in most agricultural land, particularly in sub-Saharan Africa, nitrogen and phosphor availability are most likely significant for plant growth. However, given the relation between these nutrients and the soil short-term fertilisation (e.g. manuring), we decided to not include these analyses in our study as it aims understanding and following longer-term soil fertility discussion. | Page 8 L 1-4 (Laboratory analyses): explains why these nutrients were not measured. |
| General | The future challenge is i) to concentrate crop production on the best suitable soils and ii) to improve nutrient inputs on these areas in an intensity, that yields are just water or management controlled and iii) to develop sustainable LU management techniques (e.g. conservation agriculture). This development needs help by the agriculture extension services. | This issue does not relate directly with the objectives of the current paper, which do not aim at suggesting management techniques to improve SQ. The aim is to suggest a SQ toolbox that helps to evaluate the conditions of a soil, in regard to its potential. | |
| General | The mapping of the best suitable soils should be oriented to technical knowledge for its comparability, however should include farmers views. The general objective of the paper just moves to the right direction, the presented toolbox however needs improvement (reduction in units). | Our objective is not to create a map or tools to map, it is to enable the farmers to optimise their SQ evaluation. However, for mapping purposes, each criterion used can be mapped separately, which presents the advantage to evaluate the various issues separately (organic matter availability, erosion, soil types). | |
| P3, Table 1, L pH | I suggest, that doing numerous measurements on soil pH is cheaper by application of the sensor technique instead of the Hellige test kit | | Table 1 L pH |

| Position in manuscript | Reviewer's comments | Authors' answer | Changes on new manuscript |
|---|---|---|---|
| P4, Table 2 | add a row with information, for which sort of soils (types, region) and land use the soil quality indicators were proposed | | Table 2: information added. |
| P4 L10 | Verlinden and Dayot, 2005 | | |
| P6 L31 | what is meant with color shade ? The standardized MUNSELL soil color charts are composed by the variables hue, value and chroma. Is shade identical with value ? Please explain. | Colour shade in common language would refer to colour value in the Munsell colour system. We use both to separate the perception of shade (dark or light) against the numeric evaluation of darkness (colour value). | |
| P7 L12 | "two sample rings": defined volume? calculation of bulk density possible ? please explain or reformulate | | Page 7 L. 12-13 |
| P9 Table 4 | hardness is an often mentioned indicator for soil quality. I suggest, that the hardness is related to the condition of the soil in the (almost) dry state, perhaps for that time of the year, when ploughing is done. Please add some explanations on the local farmers intention. | The consistence, or the concept of hardness, is understood under dry conditions, which impacts importantly the difficulty of ploughing (often performed as soon as possible in the season). | Page 8, L21-23 (beginning of the chapter 3 "Results and Discussion") and Caption Table reflab:4: Clarify the meaning of hardness/ consistence. |
| P10 L21 | values of pH (8.4 to 10.1) are not existing in Table 5 ! | These pH values are pH in water, while in Table 5 the values are pH in $CaCl_2$. | Page 10 L3-4 to clarify this difference. |
| P11 Table 5 | in row <20 $\mu m$ – sub data of sand are give and in row sand – sub data of <20 um are given. Check all data and compare with data in respective chapters. | It seems that some calculation errors are in the Table 5 and 7 | modify the Table 5 and 7 in order to clarify the particle size content. |
| P11 Table 5 | add row with WRB classification | WRB is not of first importance for evaluating SQ, but it can help understanding soils from an international perspective. | Table 6 added, which includes all WRB profiles' names. |
| P12 Table 6 | same mistakes as for table 5 | | modify the Table 7 in order to clarify the particle size content |

| Position in manuscript | Reviewer's comments | Authors' answer | Changes on new manuscript |
|---|---|---|---|
| P12 L6 | acc. to tab. 6 the coefficient of variation is large for TOC, moist color value, < 20 $\mu m$ fraction but not for pH (in both depth intervals CV < 0.2) | This is a good observation and we removed pH from the technical analyses that have a high CV | Page 10 L17 |
| P13 Fig2 | this graph pretends a precise depth distribution which was not analysed. Additionally this graph is redundant, please delete. | This remark is relevant | Table deleted |
| P13 L8 | fragipan: delete term because of its vague definition | | Page12 L6 replace fragipan by "hard soil layer" |
| P14 L8 | "large variety of soils" -> large variety of soil properties | accept the change to "soil properties" | Page 12 L25 |
| P14 L9 | "standardize the assessment of the SQ at a specific location and time". Why time ? Soil quality assessments always results in a potential for intended land use. Different climatic conditions may be included in the potential. Thus the results are irrespective of time, however may be altered by changes in soil properties due to land use. | SQ is not only about a potential. It represent the potential in regards to various climatic conditions, but it is the consequence of various soil degradation or improvements techniques. Therefore, it is important the notion of time. The SQ of a specific site can change though agricultural activities. | |

| Position in manuscript | Reviewer's comments | Authors' answer | Changes on new manuscript |
|---|---|---|---|
| P14 L9 | What is meant with location in this context: Three villages were studied, should the SQ assessment by different for each village? | Not in this context. The assessment would be done following the same method in all villages. What we mean is that it is important to evaluate SQ at various location and to compare the results between the locations. The comparison is important in order to evaluate the potential that can be reached in specific regions (villages), there is no need to explain that a soil in Ekolola (woodland) is bad and in Omhedi (Oshana) it is good. It is more useful to differentiate bewteen various location in a same village, to evaluate the potentials. | |
| P14 L16 | harder in dry conditions (?) | add "in dry condition" after harder | Page 14 L 4 |
| P14 L21 | this increase in $< 20\ \mu m$ can only be marginal | The increase in fine particle content can be significant by mining riverbeds for example, following researches from Kreike (2013), as explained in the manuscript (Page 14 L 9). | |
| P15 L12 | Data presented by Blume et al 2011 cannot be transferred to Namibian soils. | It is a relevant comment given the origin of the soils used in Blume 2011, difficult to compare to the Namibian context. However, we did not find similar relations adapted to tropical soils. Moreover, the results indicate a relatively well-balanced repartition of SOC status in our soils, which therefore helps to analyse the SQ status of a soil in relation to other soils of the same region. | |

| Position in manuscript | Reviewer's comments | Authors' answer | Changes on new manuscript |
|---|---|---|---|
| P16 L8 | "indicates important degradation". Relevant forms of degradation (acidification, salinization, decline in nutrients, compaction) do not include the shift to more coarse particles. | Processes that can remove fine particles from the top soil are 1) eluviation related to dispersive salts; 2) overland flow erosion, 3) wind erosion. | Page 16 L 13.Add suggestions of processes leading to soil texture coarsening (e.g. overland flows, eluviation). |
| P16 L9 | major soil improvements": see above. | The increase in fine particle content can be significant, following researches from Kreike (2013). | Page 14 L 9 |

**Answer to reviewer 2**

| Position in manuscript | Reviewer's comments | Authors' answer | Changes on new manuscript |
|---|---|---|---|
| page3 | From my point of view, it is an important paper, which could be improved significantly by using an international soil classification (WRB) and description of the different KwSUs, making it accessible for a wider audience and allow for international comparison and land management studies in other areas comprising comparable environmental conditions. | We agree with this concern. However, given the lack of diagnostic properties, the WRB is poorly informative in this context. | Page 20 Table A1 (Appendix A): Add three soil descriptions, including pictures and WRB; Page 10, L 8: Add a short chapter following "International classification". Page 13 Table 6: Added table, in which all soil profiles are classified using the WRB. Page 12 L 29-2: Comment concerning WRB results (in 3.5.1 "Importance of a soil quality evaluation toolbox"). |
| General | Photographic documentation of soil profiles (if available) and profile descriptions seem appropriate making it more attractive and better accessible to the readers. | | Page 20 Table A1 (Appendix A): Soil descriptions and pictures added. |
| P7 L23: I don't fully agree with the argument against the measurement of the Cations exchange capacity. | | Page 7 L 31-31 (§ Methods): Change this section to clarify the decision. | |
| 2.3.2 Laboratory analyses | As high contents of carbonates and salts are expected it could be important to know which kind of salts are present to be able to adapt land management. | | Page 7 L28: Added this salt types. |
| general | It is not clear how soil fertility/chemical fertility, used in results and discussion, is defined in this study: Is it the potential of the soil to provide nutrients coming from natural sources or artificial with fertilizers? Or the plant available nutrients? | We should clarify what chemical refers to... | Page 10 L1 Chemical fertility definition clrarified. |

| Position in manuscript | Reviewer's comments | Authors' answer | Changes on new manuscript |
|---|---|---|---|
| results and discussion | the authors refer to chemical fertility, I would suggest replacing chemical fertility with soil fertility, as chemical fertility includes available nutrient contents, which were not measured. | We always used "chemical fertility potential" to clearly indicate that it is not the actual chemical fertility (related to nutrient content) but an indicator for the potential that the soil could reach if sufficiently fertilised. We think that replacing "chemical fertility" by "soil fertility" will add confusion to the reader. | |
| § 3.2, page 10 | The authors suggest a high chemical fertility and chemical exchange capacity for the *omutunda* units. This is misleading since it gives the reader the feeling that this soil is highly fertile. It should be made clear that this is relatively seen. | | § 3.2, Page 10 L.1: "...the higher potential of *omutunda* to provide nutrients, coming from any sources, compared to the other KwSUs." |
| Fig. 1 | needs a reference of the satellite image and hydrology data. | | Page 6 figure 1: Add origin of the satellite images and hydrology data (caption). |
| Fig. 1 | A little box indicating the section of the study area in the map of Namibia would be useful. | | Page 6 figure 1: Add the suggested box. |
| Fig. 2 | needs some clarification as it seems that pH and <20 $\mu m$ content was measured in high resolution and vary in depth. | This figure was removed given the different depth resolution illustrated compared to the rest of the data used. | |

**Answer to Reviewer 3**

| Position in manuscript | Reviewer's comments | Authors' answer | Changes on new manuscript |
|---|---|---|---|
| | Including soil descriptions using WRB soil classification increases the relevance to the broader public. | We agree with this concern. However, given the lack of diagnostic properties, the WRB is poorly informative in this context. | Page 20 Table A1 (Appendix A): Add three soil descriptions, including pictures and WRB; Page 10, L 8: Add a short chapter following "International classification". Page 13 Table 6: Added table, in which all soil profiles are classified using the WRB. Page 12 L 29-2: Comment concerning WRB results (in 3.5.1 "Importance of a soil quality evaluation toolbox"). |
| Table 10 | The presented toolbox seems useful, but detecting very slight colour differences in the field will not be easy. | It must be emphasized that this toolbox is a suggestion that would require further developments (Page 16 L30) and Abstract L. 12. Considering the comment and the low accuracy of Munsell colour evaluation, we modified the colour shade classes defined in the toolbox in order to include more soil colour values in the neutral class (0)(Table 10). This result in more soils classified into this class, 1) avoiding an overinterpretation of changes to be undertaken and 2) corresponding better to the farmers SQ evaluation. The values are then adapted to avoid overinterpretation of field data collected. | Page 16 L30. Page 17 Table 10: Change the SQ evaluation classes of Colour shade. |

| Position in manuscript | Reviewer's comments | Authors' answer | Changes on new manuscript |
|---|---|---|---|
| methods | As in many tropical agricultural soils, fertility in terms of N and P availability will be a severe limiting factor in this area (besides water limitation). It is however not taken into account in the soil quality evaluation. I realize that it may not be a property that can readily be measured by farmers, but it should at least be discussed as an important limiting factor. | Known to be limiting nutrients in most agricultural land, particularly in sub-Saharan Africa, nitrogen and phosphor availability are most likely significant for plant growth. However, given the relation between these nutrients and the soil short-term fertilisation (e.g. manuring), we decided to not include these analyses in our study as it aims understanding and following longer-term soil fertility discussion. | Page 8 L 1-4 (Laboratory analyses): explains why these nutrients were not measured. |
| Table 1 | depth of topsoil can better be changed to soil depth or rooting depth, as depth of topsoil is defined by the user, not so much a soil property. | These properties have been copied from Wienhold et al. (2004), as suggested in the figure caption. We would therefore not change it. | |
| Table 1 | Infiltration rate, or capacity? | Water infiltration rate | Table 1. |
| P3L7 | Soil diversity: misleading term, soil variability is more apt. | | Page 3 L.2 |
| P3L14 | How do you define the process of agricultural evolution? | | Page 3 L.8: Changed into "evolution of agricultural practices". |
| P6L8 | unclear why some farmers are visited more than once, while others are not. | The farmers who showed a broad soil and agricultural knowledge during the first interview and open to discussion were visited several times. | Page 5 L.27 |
| P7L21 | Further on only pHCaCl is shown/mentioned, so why also include pHH2O here? Better remove it if you don't show further results. | pHH2O removed from Methods. | |
| P10L2 | chemical fertility is still low compared to many other soils. Differences are relative between local soils, which should be emphasized. | | Page 10 L.1. |

| Position in manuscript | Reviewer's comments | Authors' answer | Changes on new manuscript |
|---|---|---|---|
| | Also the term chemical fertility may be a bit misleading; soil fertility may be better in this context. | We used "chemical fertility potential" to clearly indicate that it is not the actual chemical fertility (related to nutrient content) but an indicator for the potential that the soil could reach if sufficiently fertilised. We think that replacing "chemical fertility" by "soil fertility" will add confusion to the reader. | |
| Fig 2 | doesn't seem to be very relevant for the story, not very comparable to the other data shown (more detailed). So I would suggest to remove it. Also values on x-axes of first and third pane are hard to understand (not in line with table above). | We would remove this figure, given the different depth resolution illustrated compared to the rest of the data used. | |
| P14L17-26 | sentences are hard to understand. Wording can be improved/clarified. | Change wording for better clarity. | Page 16 L. 12 |
| | Maybe replace "evolution" by "transition"? | | Page 16 L.11 |
| | Improvement in this context is are to follow, it seems to imply that improvement has taken place over time, but without reference in the past? What were the conditions before the improvement? | There is a lack of data to support the assumption of soil degradation or improvement. However, these processes were perceived and explained by some farmers during the interviews. | Page 18 15-17. |
| Technical/textual points | | | |
| P3L20 | "have been developed and discussed, and yielded" ... | accepted | Page 3 L.13 |
| P4L1 | "farmers and technical assessments" | "between technical and farmers assessment". | Page 3 L.15 |
| P6L2 | remove space after "Sandveld" | accepted | |
| P6L15 | insert second closing bracket after 2005. | accepted | |
| P7L26 | replace "that" by "when" | accepted | Page 7 L. 30 |
| P10L32 | "various entities" ... | accepted | Page 12 L. 16. |
| P13L5 | "meaning" | accepted | |
| P13L11 | "play an important role in fixing" | accepted | Page 6 L. 6. |